# Crystal-chemical origins of the ultrahigh conductivity of metallic delafossites

Yi Zhang [1], Fred Tutt [1], Guy N. Evans[2], Prachi Sharma [1,3], Greg Haugstad[4], Ben Kaiser[1], Justin Ramberger [1], Samuel Bayliff [3], Yu Tao [1], Mike Manno [1], Javier Garcia-Barriocanal[4], Vipul Chaturvedi [1], Rafael M. Fernandes [3], Turan Birol [1], William E. Seyfried Jr.[2] & Chris Leighton [1] ✉

Despite their highly anisotropic complex-oxidic nature, certain delafossite compounds (*e.g.*, PdCoO$_2$, PtCoO$_2$) are the most conductive oxides known, for reasons that remain poorly understood. Their room-temperature conductivity can exceed that of Au, while their low-temperature electronic mean-free-paths reach an astonishing 20 μm. It is widely accepted that these materials must be ultrapure to achieve this, although the methods for their growth (which produce only small crystals) are not typically capable of such. Here, we report a different approach to PdCoO$_2$ crystal growth, using chemical vapor transport methods to achieve order-of-magnitude gains in size, the highest structural qualities yet reported, and record residual resistivity ratios (>440). Nevertheless, detailed mass spectrometry measurements on these materials reveal that they are not ultrapure in a general sense, typically harboring 100s-of-parts-per-million impurity levels. Through quantitative crystal-chemical analyses, we resolve this apparent dichotomy, showing that the vast majority of impurities are forced to reside in the Co-O octahedral layers, leaving the conductive Pd sheets highly pure (~1 ppm impurity concentrations). These purities are shown to be in quantitative agreement with measured residual resistivities. We thus conclude that a sublattice purification mechanism is essential to the ultrahigh low-temperature conductivity and mean-free-path of metallic delafossites.

Complex oxide materials have proven to be fertile ground for the discovery of new physical phenomena and the advancement of technologically important device function, the ABO$_3$ perovskites being a prime example[1–4]. Many related classes of complex oxides offer similarly substantial potential but remain less extensively studied. ABO$_2$ delafossites are one example, where two-dimensional triangular sheets of typically monovalent A ions are separated by layers of edge-shared BO$_6$ octahedra[5–10]. Most delafossite compounds are insulating or semiconducting, such as the CuFeO$_2$ studied for frustrated magnetism[11,12] and the CuAlO$_2$ considered for transparent conductive oxide applications[13]. It has been known since 1971, however[5–8], that some delafossites exhibit metallic character, exemplified by PdCoO$_2$ and PtCoO$_2$. These metallic delafossites received little attention until the relatively recent discovery that, in bulk-single-crystal form, they are the most conductive oxides known[5,10]. While their *c*-axis resistivities (ρ$_c$) are more than 100 times higher[5,8,9,14–16], the room-temperature *a-b*-plane resistivities (ρ$_{ab}$) of PdCoO$_2$ and PtCoO$_2$ are only 3.1 and 1.8 μΩcm, respectively[17], comparable to or lower than Au[5]. Their low-temperature (*T*) residual ρ$_{ab}$ values fall as low as 8.1 nΩcm[17], implying residual resistivity ratios (RRRs) up to 376[17] and low-*T* mean-free paths

[1]Department of Chemical Engineering and Materials Science, University of Minnesota, Minneapolis, MN 55455, USA. [2]Department of Earth and Environmental Sciences, University of Minnesota, Minneapolis, MN 55455, USA. [3]School of Physics and Astronomy, University of Minnesota, Minneapolis, MN 55455, USA. [4]Characterization Facility, University of Minnesota, Minneapolis, MN 55455, USA. ✉e-mail: leighton@umn.edu

of ~20 μm[5,15]. These are astonishing values, particularly in complex oxides, in such highly anisotropic structures, and given the relatively little materials development performed.

The electronic structure of metallic delafossites is similarly noteworthy. Taking PdCoO$_2$ as an example, the established $R\bar{3}m$ structure of the 3 $R$ polymorph[5–7,9,10,18] results in a strikingly simple electronic structure where a single Pd $d$ band crosses the Fermi level, with only modest contributions from Co, and negligible O character[5,9,10,18–21]. The Pd $d$ band is highly dispersive[5,9,10,18–22], potentially assisted by a contribution from Pd $s$ states[23,24]. The triangular Pd$^{1+}$ planes thus generate a Fermi surface closely approximating a hexagonal-cross-section cylinder filling half the hexagonal Brillouin zone in the $a^*$-$b^*$ reciprocal-space plane[5,15,18–21,25,26]. The remarkable simplicity of this electronic structure is underscored by electron effective masses of only 1–1.7 $m_e$[5,15,18,27]. The edge-shared CoO$_6$ octahedra also have short Co-O bond lengths and large crystal field splitting, stabilizing low-spin ($S = 0$) Co$^{3+}$[18]. PdCoO$_2$ is thus a model metallic delafossite, which can be thought of as triangular metallic sheets of Pd$^{1+}$ separated by insulating nonmagnetic layers of Co$^{3+}$O$_6$ edge-shared octahedra.

The above attributes, particularly the exceptional low-$T$ mean-free path, and simple hexagonal Fermi surface, have enabled a remarkable string of recent achievements with bulk crystals of PtCoO$_2$ and PdCoO$_2$. These include observations of: various forms of quantum oscillation[14,15,28,29], very large positive magnetoresistance along the $c$-axis[30], potential phonon-drag effects in resistivity[15], possible hydrodynamic electron transport[28], and itinerant surface ferromagnetism[26,31,32]. Yet more recently, non-local electrodynamics[33], directional ballistic transport[34], supergeometric electron focusing[35], and non-local transport[36] have been reported in PdCoO$_2$ crystals and micron-scale structures, exploiting the long mean-free-path and anisotropic Fermi surface. The addition of magnetic moments in PdCrO$_2$ adds further richness, encompassing magnetic frustration[37–39], a complex antiferromagnetic spin structure[37–40], possible chirality[40], an anomalous Hall effect[41,42], and $T$-linear resistivity[43,44] potentially due to magneto-elastic scattering[44]. Metallic delafossites are thus making their mark in condensed matter and materials physics, and potential applications are being explored, in nanoelectronic interconnects[45], harsh-environment electronic devices[46–48], electrocatalysis[46,49], ballistic transport devices[34–36,46,47], transparent conductors[46,50], terahertz sources[51], etc.

Despite this progress, the origins of the ultrahigh conductivities and mean-free paths in metallic delafossites remain unclear. While progress has been made with understanding thermal properties of PdCoO$_2$[18] and related phonon spectra[52,53], much remains to be elucidated regarding electron-phonon scattering and coupling[5,14] and thus the low room-$T$ resistivities[5]. An important recent advance is a first-principles reproduction of the room-temperature resistivity and calculation of weak electron-phonon coupling constant in PdCoO$_2$[24]. Yet more urgently given the above advances[14,15,24,29,30,32–36], the origin of the outstanding $a$-$b$-plane residual resistivities and low-$T$ mean-free-paths remain incompletely understood[5,46]. It is widely accepted that ultrahigh purity and ultralow defect density are necessary for such behavior[5,14,15,28,54,55], but, confusingly, the crystal growth methods applied to metallic delafossites[6,14,16,56–59] are not typically capable of such[5,54]. The prevailing method for PdCoO$_2$ and PtCoO$_2$ is a metathesis-based flux growth[6,14,16,57,58], which is not well understood, typically limited to ~1-mm lateral sizes, and not performed under conditions that would be expected to realize ultrahigh purity. Multiple authors have in fact commented on the crude synthesis methods for metallic delafossites relative to other systems with comparable mean-free-paths[5,54]. On the other hand, recent electron microscopy on single-crystal PdCoO$_2$ supports sufficiently low defect densities that only upper limits could be placed[54,60]. An electronic transport study of metallic delafossites as a function of irradiation-induced point defect density also evidenced extraordinarily low initial defect densities on

the Pd/Pt planes, insensitive to the (unknown) disorder level on the B-O layers[54]. It is thus unclear what the impurity and defect densities are in single-crystal metallic delafossites, how they are divided between A and B-O planes[54], if/how defect densities are minimized by current growth methods[54], or whether some form of defect mitigation or tolerance occurs[54]. One proposal for the latter is the hidden kagome picture recently advanced to explain reduced electron-impurity scattering[61].

The current work seeks to elucidate some of the above through advances that are thus far absent from the PdCoO$_2$ literature: improvements in bulk crystal growth in terms of mechanistic understanding, crystal size, defect density, and purity; more extensive structural and chemical characterization; and comprehensive trace impurity analysis. The latter is particularly conspicuous in its absence from the metallic delafossite literature, likely because the premier method—inductively coupled plasma mass spectrometry (ICP-MS)—typically involves acid digestion. Metallic delafossites such as PdCoO$_2$ and PtCoO$_2$ are unusually stable in extreme pH, however[46–48], likely hindering prior attempts at comprehensive trace impurity analysis.

Here, we first elaborate on a different crystal growth method for PdCoO$_2$ based on chemical vapor transport (CVT), for which a simple mechanism is proposed. CVT is then compared to the established metathesis/flux method and is shown to enable orders-of-magnitude gains in crystal size and mass, the highest structural quality yet reported, and record RRR. Nevertheless, particle-induced X-ray emission (PIXE) and low-$T$ magnetometry suggest significant impurity concentrations. Enabled by microwave digestion methods, comprehensive trace impurity analyses of metathesis/flux-grown and CVT-grown PdCoO$_2$ crystals were thus performed by ICP-MS, with clear conclusions. Specifically, these metallic delafossite crystals are definitively *not* ultrapure in a general sense. Standard metathesis/flux crystals typically have ~250 ppm metals-basis impurities overall; CVT crystals are purer but still harbor ~50 ppm overall impurities. Through a detailed crystal-chemistry-based analysis, however, we show that the vast majority of these impurities must populate Co-O octahedral layers, while as few as four elements can conceivably substitute for Pd. The concentration of these elements is of order 1 ppm, ~100 times lower than the total impurity concentration, which is shown to be in quantitative agreement with the measured residual $\rho_{ab}$. We thus propose that a sublattice purification mechanism is central to the extraordinary low-$T$ conductivity of metallic delafossites, enabling ultrahigh purity in the conductive Pd/Pt sheets, largely unaffected by substantial impurity concentrations in the CoO$_6$ octahedra.

## Results

### Chemical vapor transport crystal growth

As noted in the Introduction, metathesis-based flux growth has become standard for single-crystal PdCoO$_2$ and PtCoO$_2$[6,14,16,57,58]. This is based on metathesis reactions such as PdCl$_2$ + Pd + Co$_3$O$_4$ → 2PdCoO$_2$ + CoCl$_2$, *i.e.*, PdCl$_2$, Pd, and Co$_3$O$_4$ reagents (99.995–99.999% purity in our case) reacting to form the target PdCoO$_2$ plus a CoCl$_2$ by-product, which is chemically removed (see Methods for more details). In the current study, dwell temperatures of 700–750 °C were employed for metathesis growth, followed by cooling to 400 °C at 40–60 °C/h. As shown in the scanning electron microscopy (SEM) image in Fig. 1b, the resulting product is a mass of PdCoO$_2$ crystals and crystallites. The optical microscope image in the inset to Fig. 1b highlights one of the largest crystals from a typical growth, measuring $0.8 \times 0.6$ mm$^2$ laterally, and 0.017 mm thick. This corresponds to 0.008 mm$^3$ and 0.065 mg, comparable to literature reports[16,28,34,54,55]. Typical powder X-ray diffraction (PXRD) data from such crystals are shown in Fig. 1d (green), displaying a good match with a PdCoO$_2$ standard (gray), and yielding lattice parameters $a = 2.8308 \pm 0.0002$ Å and $c = 17.747 \pm 0.002$ Å, in agreement with accepted values[5,6].

 

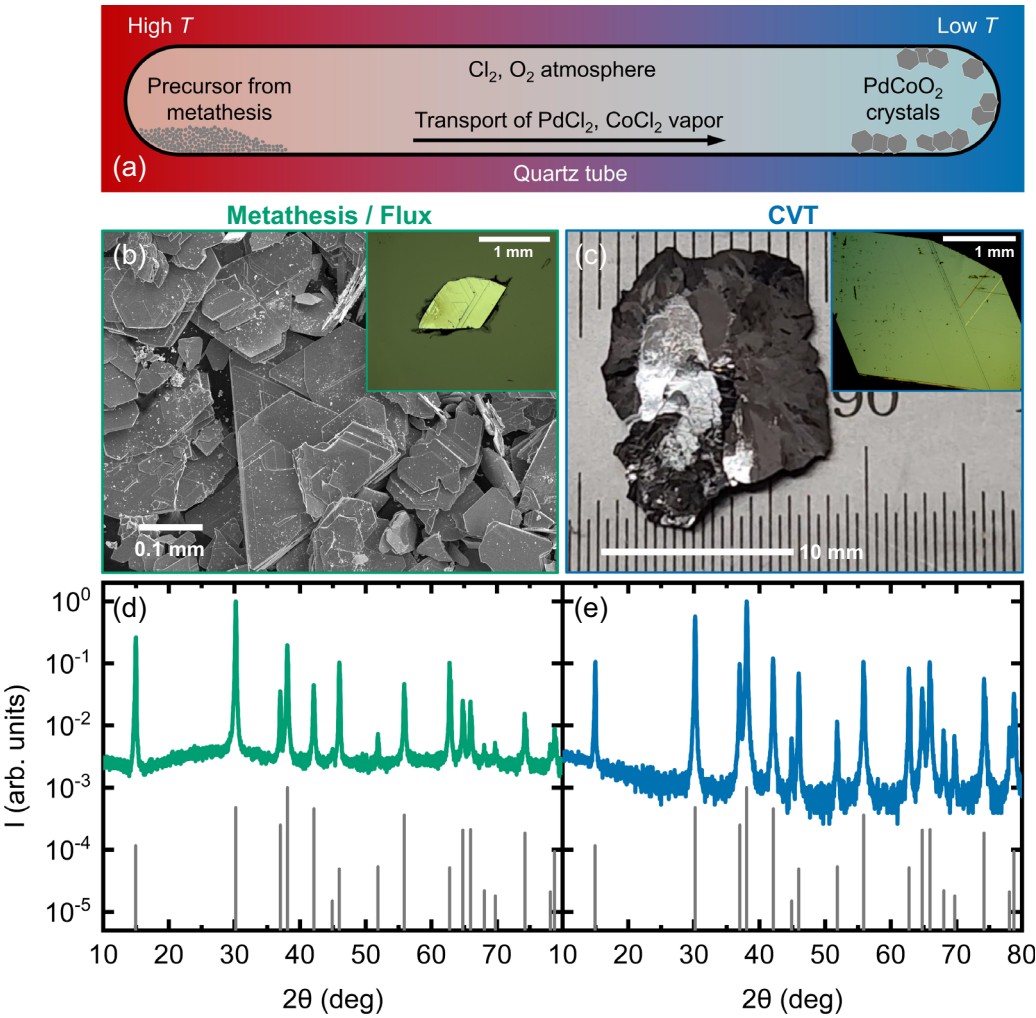

**Fig. 1 | Crystal growth, microscopy, and powder X-ray diffraction for metathesis/flux and chemical vapor transport (CVT) crystals. a** Schematic of CVT growth of PdCoO$_2$. **b** Scanning electron microscopy image of the products of metathesis/flux growth. Inset: Optical microscopy image of a larger single crystal from metathesis/flux growth (0.8 × 0.6 × 0.017 mm$^3$). **c** Optical microscopy image of a multicrystal from CVT growth. Inset: Optical microscopy image of a typical single crystal from CVT (3.5 × 2.0 × 0.17 mm$^3$). **d, e** Powder X-ray diffraction patterns (intensity *vs.* 2θ angle) from representative ground crystals from the metathesis/flux ((**d**), green) and CVT ((**e**), blue) methods; reference patterns are shown in gray.

While such crystals have enabled exciting advances[14,15,17,25–30,32–36], as discussed in the Introduction, it is nevertheless true that little progress in PdCoO$_2$ crystal size or quality has been reported since *ca.* 2007. There is also limited understanding of the growth mechanism. Most publications describe the mechanism as "flux" or "self-flux" and employ specific temperature-time trajectories but with little explicit justification[14,16,56–58]. Seeking improvement over this, we explored CVT crystal growth of PdCoO$_2$. CVT is not commonly applied to complex oxides but the fact that PdCoO$_2$ decomposes (at~800 °C in air, for example[48]) hints at CVT as a possibility, as does the relatively low decomposition temperature of PdCl$_2$ (~600 °C in low Cl$_2$ partial pressure[62,63]), a potential transport agent. CVT was thus performed in a set-up shown schematically in Fig. 1a, and elaborated upon in Methods and Supplementary Notes 1 and 2 (Supplementary Figs. 1–3 and Supplementary Tables 1 and 2). The solid precursors are coarse PdCoO$_2$ powder derived from the above metathesis growth, along with 99.999%-pure PdCl$_2$. At typical hot and cold-zone temperatures of 760 and 710 °C, we propose that PdCl$_2$ decomposition is followed by a reaction such as PdCoO$_2$(s) + 2Cl$_2$(g) ↔ PdCl$_2$(g) + CoCl$_2$(g) + O$_2$(g), *i.e.*, Pd chloride and Co chloride vapor transport across the temperature gradient in a Cl$_2$(g) and O$_2$(g) atmosphere. The reaction runs to the left in the growth (cold) zone, resulting in PdCoO$_2$ crystals

and chloride by-products, which are chemically removed (see Methods).

As shown in Fig. 1c, after optimization of hot and cold-zone temperatures, relatively large PdCoO$_2$ crystals and multicrystals derive from this process, at high yield (>90%). Specifically, the reaction product is a mix of single crystals and larger multicrystals, the latter likely forming due to nucleation of Pd or PdO seeds during the initial inverted-temperature-gradient period of the growth (see "Methods" section and Supplementary Note 1). Multicrystals up to 12 × 12 mm$^2$ laterally and 0.3 mm thick are grown (Fig. 1c), corresponding to 43.2 mm$^3$ and 345 mg. Gently breaking apart such multicrystals isolates single crystals up to 6.0 × 4.0 mm$^2$ laterally and 0.17 mm thick, corresponding to 4.08 mm$^3$ and 32.6 mg. A representative example is shown in the inset to Fig. 1c. These CVT PdCoO$_2$ multicrystals are thus larger than our metathesis/flux crystals by factors of~15 in maximum lateral dimension,~18 in thickness, and~5300 in volume and mass. Correspondingly, the CVT single crystals are larger than metathesis/flux single crystals by factors of~8 in maximum lateral dimension,~10 in thickness, and~500 in volume and mass.

As illustrated in Fig. 1e, PXRD patterns from powdered CVT crystals are also consistent with single-phase PdCoO$_2$, with $a = 2.8306 \pm 0.0002$ Å and $c = 17.746 \pm 0.002$ Å. Intriguingly, large-scale

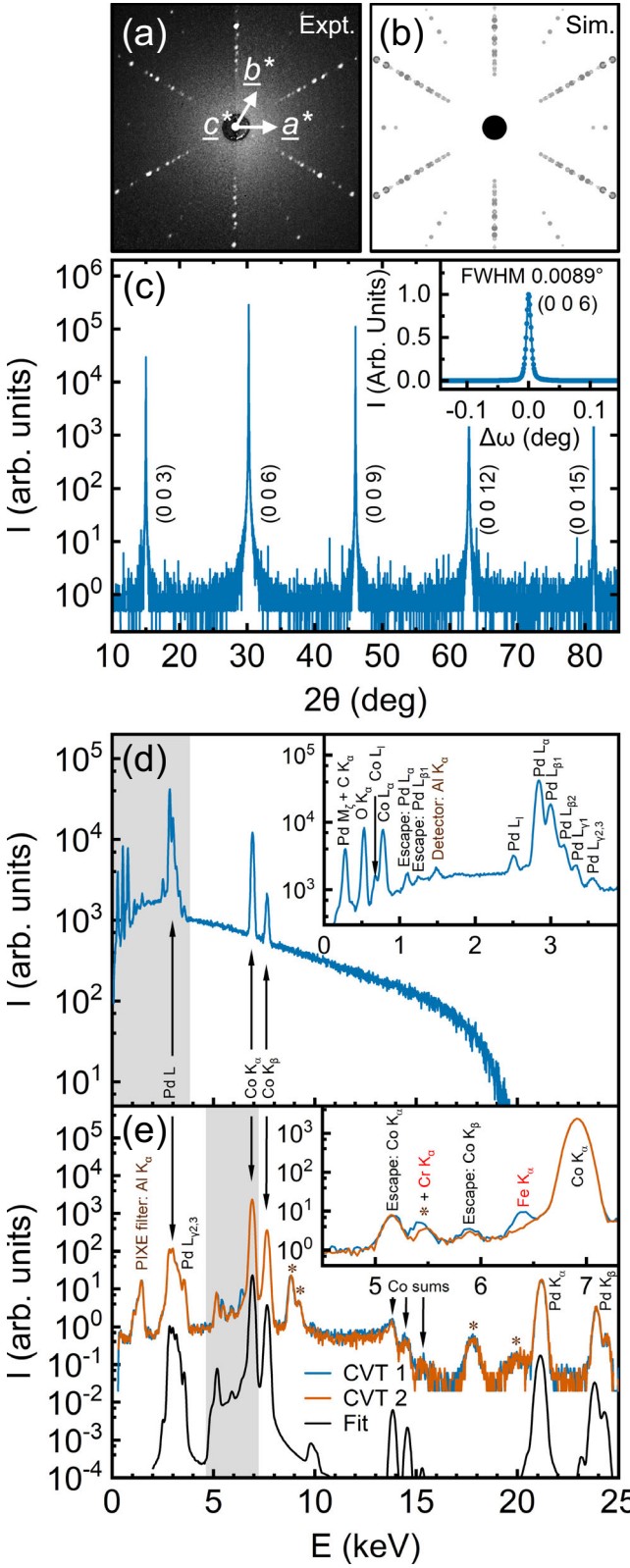

**Fig. 2 | Structural and chemical characterization of CVT-grown single-crystal PdCoO$_2$. a** Back-reflection Laue X-ray diffraction pattern along a [001] zone axis of a representative single crystal, with reciprocal-space axes labeled. **b** Corresponding simulated Laue pattern where the size of the points indicates their relative intensity and the central black circle represents the beam aperture. The simulation is based on the accepted PdCoO$_2$ structure and lattice parameters[6,79]. **c** Specular high-resolution X-ray diffraction (intensity *vs.* 2θ angle) from a representative 001-oriented single crystal, showing only 00*l* reflections. Inset: Rocking curve through the 006 reflections in (**c**), with a full-width-at-half-maximum (FWHM) of 0.0089°. **d** Energy-dispersive X-ray spectrum (intensity *vs.* energy) from a representative region (25 × 25 μm$^2$) of a single crystal. The peaks are labeled and the inset is a close-up of the 0 to 3.9 keV region shaded gray. **e** Particle-induced X-ray emission spectra from two crystals (CVT 1 in blue and CVT 2 in orange), along with the GUPIXWIN simulation result discussed in the text (black, vertically offset for clarity). The peaks labeled with an "*" are known instrumental artifacts in our ion beam accelerator/PIXE system. The inset is a close-up of the 4.5–7.3 keV region shaded gray; note the different intensities of the Cr and Fe K$_\alpha$ peaks in the two crystals. The Cr peak at 5.4 keV has partial overlap with an artifact peak but the variation in intensity from crystal to crystal strongly suggests a contribution from Cr. The Fe peak at 6.4 keV has no such overlap and also varies from crystal to crystal.

parameters are lower is in fact supported by *t*-testing, which indicates null-hypothesis *p* values below 0.0001 for both *a* and *c*, far below often-used confidence cutoffs such as *p* = 0.01. The smaller standard deviation indicates better reproducibility with CVT, while decreased cell volume often indicates lower defect density[64–69], including in complex oxides[65–69]. Minimization of unit cell volume has in fact emerged as a guiding principle for the synthesis of highly perfect complex oxides with minimized densities of non-stoichiometry-accommodating defects[65–69]. These data thus hint at lower defect densities in CVT-grown crystals, a conclusion that is reinforced below. As a final comment on structure, note that powdered samples of these CVT crystals were recently used in a 12-1000 K structural study *via* PXRD and accompanying refinement, confirming their single-phase nature and the thermal stability of the *R*$\bar{3}$*m* structure[18].

## Structural and chemical characterization

With CVT growth established to provide substantial improvements in PdCoO$_2$ crystal size and mass, and with improved defect density suggested by the lattice parameters, it is important to more rigorously assess structural quality and purity. Preliminary prior characterization established single crystallinity *via* two-dimensional single-crystal X-ray diffraction[18]. In Fig. 2a this is improved on by presenting Laue diffraction data conclusively confirming single crystallinity. The six-fold-symmetric pattern of diffraction spots down this [001] zone axis, interspersed with a second six-fold pattern of weaker spots (seen upon close examination of Fig. 2a), is as expected, as illustrated by the simulated pattern in Fig. 2b based on the accepted *R*$\bar{3}$*m* structure and lattice parameters[6]. Wide-range specular high-resolution X-ray diffraction (HRXRD) was also performed on 001-oriented crystals (Fig. 2c), showing only sharp 003 through 0015 reflections, further confirming single crystallinity and phase purity. Of particular significance, a HRXRD rocking curve through the 006 reflection is shown in the inset to Fig. 2c, revealing a full-width-at-half-maximum of only 0.0089°. To the best of our knowledge, this is the lowest rocking curve width reported for a metallic delafossite, demonstrating very low mosaic spread, another strong indicator of low defect density.

Turning to chemical characterization, Fig. 2d shows an energy-dispersive X-ray spectroscopy (EDS) scan from a representative PdCoO$_2$ CVT crystal. Only Pd *L* and Co *K* peaks are clearly visible in this wide-energy-range scan, their relative intensities giving a Pd/Co ratio of 0.98 ± 0.05, consistent with the nominal stoichiometry. The inset to Fig. 2d focuses on the low-energy region shaded gray in the main panel, revealing, with the exception of a small detector-artifact peak and the

comparisons of the lattice parameters for flux/metathesis and CVT crystals suggest subtle but significant differences. Averaged overall measured crystal batches (*n* = 26 for metathesis/flux and *n* = 18 for CVT), our metathesis flux crystals have *a* = 2.83088 ± 0.00078 Å, *c* = 17.7495 ± 0.0047 Å (where the uncertainties are now standard deviations), while CVT crystals have *a* = 2.83013 ± 0.00023 Å, *c* = 17.7446 ± 0.0018 Å. Both the lattice parameters and standard deviations are therefore lower in CVT crystals. That the lattice

inevitable C contamination in SEM/EDS, only peaks from Pd, Co, and O. Unsurprisingly, EDS is thus not sufficiently sensitive to detect any trace impurities in these $PdCoO_2$ crystals, which was the motivation for the PIXE data in Fig. 2e. PIXE is analogous to EDS except that the excitation is achieved with MeV-range He ions rather than electrons, vastly decreasing the broad background due to bremsstrahlung. As can be seen in Fig. 2e, which shows spectra from two different CVT crystals (orange and blue), PIXE thus exposes a plethora of Pd, Co, and sum peaks, at a high signal-to-noise ratio. Aside from some instrumentation artifacts (labeled with asterisks in Fig. 2e), almost all aspects of these spectra are reproduced by a GUPIXWIN[70] simulation for $PdCoO_2$, shown as the black line. The inset to Fig. 2e, however, shows a close-up of the 4.5–7.3 keV region (shaded gray in the main panel), highlighting two peaks that are not accounted for by Pd, Co, or O. These peaks also exhibit crystal-to-crystal variations in intensity, strongly suggesting that they derive from trace impurities. The ~5.4 keV peak very likely indicates Cr impurities, while the 6.4 keV peak is very likely due to Fe. Full quantification of impurity concentrations at this level is challenging for PIXE but our best estimates based on GUPIXWIN fits to Pd $L$, Co $K$, and Fe $K$ peaks suggest Fe concentrations of $\gtrsim 50$ ppm. (As for all impurity concentrations in this paper, this value is on a weight basis, *i.e.*, it is in µg/g). While this is only approximate, this is our first indication that even CVT-grown $PdCoO_2$ crystals may not actually be ultrapure. As a final comment on these PIXE data, we note that Cl was not detected; the estimated limit of detection is of order 1000 ppm, however, due to the overlap of Cl $K$ and Pd $L$ peaks.

## Electronic and magnetic properties

Electronic transport is an excellent relative probe of total defect and impurity densities, and thus $T$-dependent $\rho_{ab}$ measurements were made on our CVT-grown crystals. It is essential to acknowledge at this point that such measurements are nontrivial for single-crystal $PdCoO_2$[17,28,54,57], because of the extremely small low-$T$ resistances (due to the very low residual $\rho_{ab}$), and the sizable $c$-axis/$a$-$b$ plane resistivity anisotropy (~350–750, depending on $T$[8,14–16]). The most accurate measurements in the literature in fact use focused-ion beam techniques to pattern structures with long channel lengths to increase resistance, at the same time fully constraining the current path in the $a$-$b$ plane[17,28,54,57]. In our case, the availability of larger crystals than in prior work enabled cleavage into relatively long and thin bar-shaped samples for simpler, in-line, top-contact, four-wire measurements. The inset to Fig. 3a shows an example of such an in-line geometry on a 1.2-mm-long, 23-µm-thick crystal. Even in this case, residual resistances were found to be~20 µΩ, necessitating low-noise measurements and careful accounting for instrumental offset voltages. Full details are provided in Methods and Supplementary Note 3 (including Supplementary Figs. 4–6 and Supplementary Tables 3–5). Briefly, an AC resistance bridge with a pre-amplifying channel scanner was employed, using parallel measurements on superconducting V thin films to determine and account for offsets. Top-contact geometries such as those shown in the inset to Fig. 3a are also subject to systematic error due to the large $\rho_c/\rho_{ab}$[8,14–16]. As described in Supplementary Note 3, finite-element simulations of current flow and potential drop were thus performed in our specific measurement geometry. These confirm overestimation of $\rho_{ab}$ due to the large $\rho_c$ that generates potential drop through the crystal thickness. This issue worsens with cooling (as $\rho_c/\rho_{ab}$ increases), meaning, importantly, that RRR values in this geometry represent lower bounds on true values (see Supplementary Note 3 for details).

Based on the above procedures, presented in Fig. 3a is $\rho_{ab}(T)$ for the representative CVT-grown $PdCoO_2$ crystal shown in the inset;

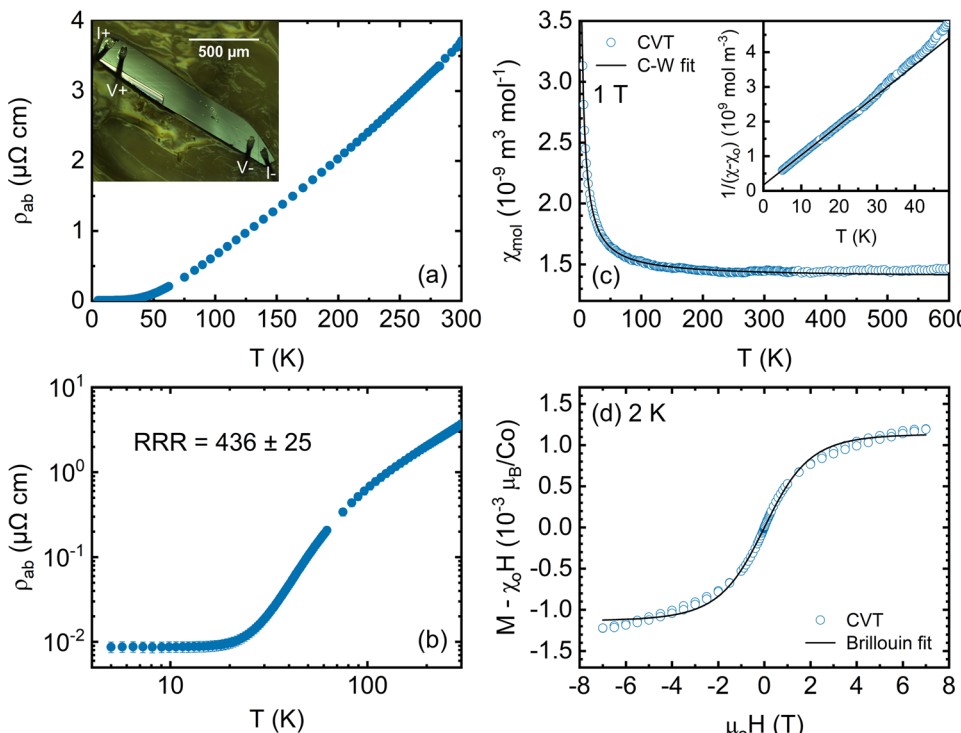

**Fig. 3 | Electronic and magnetic properties of CVT-grown single-crystal $PdCoO_2$. a** a-b-plane resistivity ($\rho_{ab}$) *vs.* temperature ($T$) on a representative single crystal. Inset: Image of the crystal and four-point in-line contact arrangement. **b** Data from (**a**) on a $\log_{10}$-$\log_{10}$ scale, highlighting the residual resistivity ratio of $436 \pm 25$. The error bars in (**b**) represent the uncertainty estimated from offset subtraction (dominant at low-$T$) and sample dimension measurements. **c** Molar magnetic susceptibility ($\chi_{mol}$) *vs.* $T$ in a 1 T magnetic field (blue open circles), from a representative powdered crystal. The black solid line is a fit to a sum of Pauli and Curie-Weiss susceptibilities, as discussed in the text. Inset: $T$ dependence of ($\chi_{mol}$ - $\chi_0$)$^{-1}$, where $\chi_0$ is the $T$-independent susceptibility. The solid black line is a 5–20 K Curie-Weiss fit. **d** 2-K magnetization ($M$, corrected for $\chi_0$) *vs.* magnetic field ($\mu_0 H$) for the crystal in (**c**) (blue open circles). The solid black line is a Brillouin fit described in the text.

Fig. 3b shows the same data on a $\log_{10}$-$\log_{10}$ scale. The 300 K resistivity of this representative crystal is 3.65 μΩcm, which can be compared to reported values of 2.0–6.9 μΩcm using standard contact methods[8,14–16] and 3.1 μΩcm using more accurate focused-ion-beam-patterned geometries[17]. Our value of 3.65 μΩcm is thus indeed slightly larger than the most accurate determination of 3.1 μΩcm[17], but within the bounds expected for overestimation of $\rho_{ab}$ due to the top-contact geometry (see Supplementary Note 3). On cooling, the $\rho_{ab}(T)$ of this crystal saturates below 10–20 K, reaching a measured value of 8 nΩ cm. Taking this at face value, and considering the residual resistivity measurement uncertainty (dominated by offset subtraction), a RRR of $436 \pm 25$ is indicated. As noted above and discussed in detail in Supplementary Note 3, however, this RRR represents a lower bound due to the top-contact geometry. Finite-element simulations suggest a true residual $\rho_{ab}$ of ~ 4.4 nΩ cm, yielding an actual RRR up to ~670 (see Supplementary Note 3). Even without this correction, however, our measured RRR of ~440 is a record in PdCoO$_2$, which can be compared to 376 in prior metathesis/flux-grown crystals measured with focused-ion-beam patterning methods[17] (or 407 based on less accurate contacting methods[14]). In addition to realizing order-of-magnitude gains in crystal size and mass, our CVT approach thus also results in measurably lower disorder in electronic transport.

Magnetometry measurements provide another probe of crystal quality and impurity concentrations in PdCoO$_2$ and were thus performed on CVT crystals. A typical result for the $T$-dependent molar susceptibility ($\chi_{mol}$) is shown in Fig. 3c. As illustrated by the black solid line, these data are well described by $\chi_{mol}(T) = \chi_0 + C/(T - \theta)$, i.e., a sum of $T$-independent Pauli-related and $T$-dependent Curie-Weiss contributions, where $C$ is the Curie constant and $\theta$ is the Curie-Weiss temperature. The Pauli contribution derives from delocalized Pd $d$ electrons, the $\chi_{Pauli} = \frac{3}{2}\chi_0 = 1.65 \times 10^{-4}$ emu mol$^{-1}$ Oe$^{-1}$ (the factor of 3/2 accounts for Landau diamagnetism) in this case being near the middle of the distribution of literature values ($0.9 \times 10^{-4}$ to $2.9 \times 10^{-4}$ emu mol$^{-1}$ Oe$^{-1}$, see Table 1). The low-$T$ Curie-Weiss tail, however, which should not ideally occur in PdCoO$_2$ (the Co$^{3+}$ ions are low-spin, $S = 0$), very likely arises from local moments due to magnetic impurities. As shown in Table 1, the $\theta = -2.6$ K extracted from the fit in Fig. 3c is indeed small, indicating weak inter-moment interactions, as would be expected of dilute impurities. This is reinforced by the inset to Fig. 3c, where $(\chi_{mol} - \chi_0)^{-1}$ is plotted $vs.$ $T$, the 5–20 K fit yielding $\theta = -2.0$ K.

Interestingly, the Curie constant extracted from the fit in Fig. 3c yields an effective number of Bohr magnetons ($\mu_{eff}$) as high as 0.09 $\mu_B$ per formula unit (Table 1). Further insight into this is provided in Fig. 3d, which shows the applied magnetic field ($H$) dependence of the

### Table 1 | Summary of parameters from analysis of magnetometry data on PdCoO$_2$ crystals in this work and in prior literature

| Reference | Measurement details | $\chi_0$ (emu mol$^{-1}$ Oe$^{-1}$) | $\theta$ (K) | $\mu_{eff}$ ($\mu_B$/formula unit) |
|---|---|---|---|---|
| Tanaka et al. [16] | Ground crystal | $1.9 \times 10^{-4}$ | −3.50 | 0.66 |
| Tanaka et al. [16] | $H \perp c$ | $1.1 \times 10^{-4}$ | 10.63 | 0.69 |
| Tanaka et al. [16] | $H \parallel c$ | $1.6 \times 10^{-4}$ | 0.69 | 0.57 |
| Tanaka et al. [58] | Ground crystal | $0.6 \times 10^{-4}$ | −12.13 | 0.78 |
| Takatsu et al. [14] | $H \perp c$ | $1.9 \times 10^{-4}$ | −38.9 | 0.20 |
| Takatsu et al. [14] | $H \parallel c$ | $1.7 \times 10^{-4}$ | −40.8 | 0.15 |
| This work, metathesis/ flux | Ground crystal | $2.0 \times 10^{-4}$ | −8.1 | 0.37 |
| This work, CVT | Ground crystal | $1.1 \times 10^{-4}$ | −2.6 | 0.09 |

Shown are the literature reference, measurement details, temperature-independent susceptibility ($\chi_0$), Curie-Weiss temperature ($\theta$), and effective number of Bohr magnetons from the low-temperature Curie-Weiss contribution ($\mu_{eff}$). Note that the Pauli susceptibility can be calculated using $\chi_{Pauli} = \frac{3}{2}\chi_0$, as discussed in the text.

magnetization ($M$) at 2 K, after correcting for $\chi_0$. The solid black line Brillouin fit captures the data moderately well, yielding a reasonable $J = 1.2$, along with a concentration of local moments corresponding to ~130 ppm. (This is again on a weight basis; the atomic mass of Fe was assumed to convert from a number density of impurities (from the fit in Fig. 3d) to a weight-basis concentration). Taking this $J$ and combining it with the Curie constants from the fits in Fig. 3c then yields local moment concentrations of ~200 ppm. The low-$T$ paramagnetism in Fig. 3c, d, can thus be consistently described by dilute, non-interacting local atomic moments, at concentrations of order 100 ppm. While some fraction of this Curie-tail magnetism no doubt arises from surface contamination, this is a second indication, along with PIXE data (Fig. 2e), that even CVT-grown PdCoO$_2$ crystals host substantial impurity concentrations. Nevertheless, the $\mu_{eff}$ values from the Curie-Weiss tails of CVT-grown crystals do represent a significant improvement over metathesis/flux crystals. This is emphasized in Table 1, which includes data from our metathesis/flux crystals, in addition to prior literature[14,16,58]. The Curie-Weiss $\mu_{eff}$ of our CVT crystals is over 4 times lower than our metathesis/flux crystals and is the lowest reported for PdCoO$_2$. Literature values on metathesis/flux-grown crystals in fact span 0.15–0.78 $\mu_B$ per formula unit (Table 1)[14,16,58], implying substantial magnetic impurity concentrations, even in crystals with large RRR[14]. This apparent contradiction−clear indications of significant densities of magnetic impurities from magnetometry, in crystals that exhibit low disorder in transport−appears not to have been addressed in the literature. While CVT crystals are improved in this regard (they have the lowest $\mu_{eff}$), and surface contributions are likely, the implied level of magnetic impurities is concerning. For this reason, and to augment the chemical characterization in Fig. 2d, e, we thus undertook trace impurity analysis $via$ ICP-MS.

### Impurity analysis

As alluded to in the Introduction, ICP-MS-based trace impurity analysis on PdCoO$_2$ is complicated by the extraordinary acid etch resistance of metallic delafossites[46–48], which hinders simple acid digestion. We thus employed microwave digestion techniques (see Methods for details), achieving complete digestion in ~15 bar of aqua regia or concentrated nitric acid vapor at 200 °C. ICP-MS was then performed on both metathesis/flux and CVT PdCoO$_2$, quantifying the concentrations of 54 selected elements in a metals-basis approach (see Methods and Supplementary Note 4). This led to the immediate conclusion, contrary to the naïve view, that PdCoO$_2$ crystals are $not$ generally ultrapure (Table 2). Averaged over multiple crystals and ICP-MS runs, the total metals-basis impurity concentration in our metathesis/flux crystals is in fact $251 \pm 3$ ppm (Table 2). We emphasize that this value derives from the same synthesis method used in many prior reports on PdCoO$_2$[5–8,14–17,25–30,32–36,54,55,58,60] with $\rho_{ab}$ down to 8 nΩ cm and RRR up to 376[17]. Our CVT crystals are purer, as might be expected of vapor-based growth, but still retain an average overall metals-basis impurity concentration of $47 \pm 8$ ppm (Table 2).

Such substantial impurity concentrations, which are broadly consistent with PIXE and magnetometry (Figs. 2e and 3c, d), frame a central question for the remainder of this work: How can the record residual $\rho_{ab}$ of these materials, and their exceptional RRR, be reconciled with such impurity concentrations? To underscore the importance of this question, note that the reported 20-μm low-$T$ mean-free path of PdCoO$_2$ would require impurity concentrations several orders-of-magnitude lower than the above ICP-MS values, a point that we return to quantitatively below. Analysis of the distribution of the detected impurity elements casts much light on this issue. To this end, we first elaborate a simple classification scheme for the various substitutional elemental impurities that can be accommodated in PdCoO$_2$. Figure 4a depicts the crystal structure of PdCoO$_2$, along with a classification for potential A- and B-site impurities in the ABO$_2$ delafossite structure. The critical point here is that, very unlike ABO$_3$ perovskites

**Table 2 | Summary of average impurity concentrations (in weight-basis ppm) in metathesis/flux- and CVT-grown PdCoO$_2$ crystals, as determined by ICP-MS**

| Total Impurities in PdCoO$_2$ | | | | | |
|---|---|---|---|---|---|
| | Known A-site ABO$_2$, fits Pd site PdCoO$_2$ (µg/g) | Known A-site ABO$_2$, does not fit Pd site PdCoO$_2$ (µg/g) | Fits Co site, PdCoO$_2$ (µg/g) | Known B site ABO$_2$, does not fit Co site PdCoO$_2$ (µg/g) | All other (µg/g) | Total (µg/g) |
| Flux | 4.41 ± 0.03 | 21 ± 2 | 132 ± 2 | 67 ± 1 | 27.3 ± 0.5 | 251 ± 3 |
| CVT | 1.9 ± 0.3 | 0.4 ± 0.1 | 42 ± 7 | 0.6 ± 0.2 | 1.9 ± 0.6 | 47 ± 8 |

The averages are over 2–3 runs on 1–3 crystals, and the uncertainties listed are standard deviations. Na, B, and Si, which are common glassware impurities, were excluded from all analyses. First column: Total concentrations of elements known to adopt the A-site in ABO$_2$ compounds that also fit the Pd site in PdCoO$_2$. Second column: Total concentrations of elements known to adopt the A-site in ABO$_2$ compounds that do not fit the Pd site in PdCoO$_2$. Third column: Total concentrations of elements that fit the Co site in PdCoO$_2$. Fourth column: Total concentrations of elements known to adopt the B-site in ABO$_2$ compounds that do not fit the Co site in PdCoO$_2$. Fifth column: Concentration of all other elements measured (*i.e.*, those not in columns 1–4). Sixth column: Total of all prior columns.

for example, there is a massive asymmetry in the number of elements that can conceivably populate A *vs.* B sites. This arises because the B-site in metallic ABO$_2$ delafossites involves a common cation valence in oxides (3+) and a common coordination with O$^{2-}$ (octahedral); this is as in perovskites, which can accommodate much of the periodic table on the B site. The A-site in metallic ABO$_2$ delafossites, on the other hand, involves a relatively uncommon valence in oxides (1+) and a far less common coordination with O$^{2-}$ ions (linear, see the structure in Fig. 4a). This situation, which contrasts starkly with ABO$_3$ perovskites, greatly reduces the possible accommodation of A-site *vs.* B-site substitutional impurities in ABO$_2$ compounds.

Considering this in more detail, as shown in Fig. 4a, potential A-site-substituting impurities can be classified in terms of elements that are known to adopt the A-site in ABO$_2$ compounds, that either can (dark yellow) or cannot (light yellow) feasibly substitute for Pd in PdCoO$_2$. We determine this feasibility by applying Goldschmidt-type criteria, requiring <30% ionic size mismatch, an ability to adopt the required coordination, and valence within ±1 of the host element. Figure 4b then shows a periodic table of select elements most relevant to ABO$_2$ compounds, labeled with their valence, coordination number, and ionic radii in the state most closely matching the delafossite structure. As shown in the figure, Li$^+$, Na$^+$, K$^+$, Rb$^+$, Cu$^+$, Ag$^+$, Pt$^+$, and Hg$^{2+}$ (shaded yellow) are known to adopt the A-site in ABO$_2$ compounds, but in most cases cannot feasibly substitute for Pd in PdCoO$_2$. In particular, Li$^+$, Na$^+$, K$^+$, and Rb$^+$ (light yellow) only form ABO$_2$ compounds with distinctly different structure to delafossite PdCoO$_2$, with large coordination number with O$^{2-}$. In most cases their ionic radii in this coordination are also far beyond the 30% mismatch for feasible substitution for Pd. In fact, the elements that can feasibly substitute for Pd in PdCoO$_2$ (dark yellow) number only four: Cu, Ag, Pt, and Hg, highlighted via the bold border in Fig. 4b. Hg substitution would also be a special case, as the 2+ valence (Hg does not typically adopt 1+ valence in 2-fold coordination) would require an additional charge-balancing defect. This general concept is supported by prior work on deliberately Mg-substituted PdCoO$_2$, where A-site substitution with Mg$^{2+}$ was argued against based on similar reasoning[71].

The situation on the B site is very different. As shown in Fig. 4a, potential B site substituents can be classified in terms of those that are known to adopt the B site in delafossites and can feasibly substitute for Co (darkest blue), those that are known to adopt the B site in ABO$_2$ compounds but cannot feasibly substitute for Co (lightest blue), and those for which it is unknown whether they form ABO$_2$ compounds but do fit the Co site in PdCoO$_2$ (intermediate blue). Again, feasibility is based on <30% size mismatch, 3 ± 1 valence, and octahedral coordination. As shown in Fig. 4b, the total group of possible B-site substituents is numerous, totaling at least 40 (all blue shades), remaining as high as 20 even after applying our fit criteria (darkest blue shades). In contrast to ABO$_3$ perovskites, the fundamental crystal chemistry of the ABO$_2$ delafossites thus results in a large asymmetry between the number of elements that can exist as substitutional impurities on the A and B sites.

Figure 4c next maps the measured average concentrations of various ICP-MS-detected elements in CVT-grown PdCoO$_2$ crystals onto the classification scheme in Fig. 4a, b. This periodic table employs the log$_{10}$ color scale of average impurity concentrations shown to the right, also labeling the concentration and limit of detection beneath each element. The conclusion is simple. Substantial concentrations are found of elements that are likely to populate the Co site in PdCoO$_2$ (*e.g.*, Mn, Fe, Ni, Al, Cr, Ir, Sn; the darker blue shades in Fig. 4b), but not of elements that can feasibly populate the Pd site. Neither Cu, Ag, Pt, or Hg (in the dark border in Fig. 4b, c) reach even 1 ppm in CVT crystals. These findings are further quantified in Tables 2–4, where Table 2 summarizes the total impurity concentrations of the elements likely to populate the Pd and Co sites in PdCoO$_2$. In metathesis/flux crystals, of the total impurity concentration of 251 ppm (far right, Table 2), at least 132 ppm are elements that would be expected to populate the Co site (darkest blue shades in Fig. 4b), whereas only 4.4 ppm are elements that could feasibly populate the Pd site (Cu, Ag, Pt, Hg, dark yellow in Fig. 4b). We thus define a sublattice purification ratio of 4.4/251 = 1.8% of the total impurity concentration that could feasibly substitute for Pd. As also shown in Table 2, the equivalent for CVT PdCoO$_2$ is 1.9/47 = 4.0% of the total average impurity concentration that could feasibly substitute for Pd, in this case only 1.9 ppm.

Tables 3 and 4 break these concentrations down further, into the only four elements that can feasibly populate the Pd site (Table 3) and the most prevalent elements likely to populate the Co site (Table 4). The former group consists of only Ag, Hg, Cu, and Pt, while the latter group includes Al, Cr, Mn, Fe, and Ni. Notable here is that one of the largest differences between CVT and metathesis/flux crystals is the decrease in Al, possibly due to the high thermal stability and low volatility of Al oxides. Another difference is the non-negligible concentration of impurities in metathesis/flux crystals that are not obviously capable of populating either the A or B sites. Surface contamination, strained B-site accommodation, and uptake in the minor impurity phases sometimes detected in metathesis/flux crystals (see Supplementary Fig. 3a and associated discussion) can likely account for this. The concentrations of Cr, Mn, Fe, and Ni are also of particular interest, due to their likely magnetic character, no doubt contributing to the Curie-Weiss susceptibility in Fig. 3c, d, and Table 1. Quantitatively, the estimated magnetic impurity concentration from Table 4 is 31 ppm in CVT crystals, compared to order 100 ppm from magnetometry. This is reasonable agreement considering the likelihood of additional surface contamination in magnetometry. In terms of comparisons to PIXE, ICP-MS on CVT crystals indicates 9 ± 2 ppm of Fe, also comparing reasonably to the order-50-ppm approximate estimate from PIXE (Fig. 2e). Finally, ICP-MS was also used to determine the Pd/Co ratio of CVT crystals, yielding 1.08 ± 0.02. This is inconsistent with the EDS result of 0.98 ± 0.05 at first sight, but EDS data were acquired from 25 × 25 µm$^2$ regions free of surface particles. These particles are of course not excluded from ICP-MS, and thus remnant surface Pd and PdCl$_2$ likely skews the ICP-MS Pd/Co ratio.

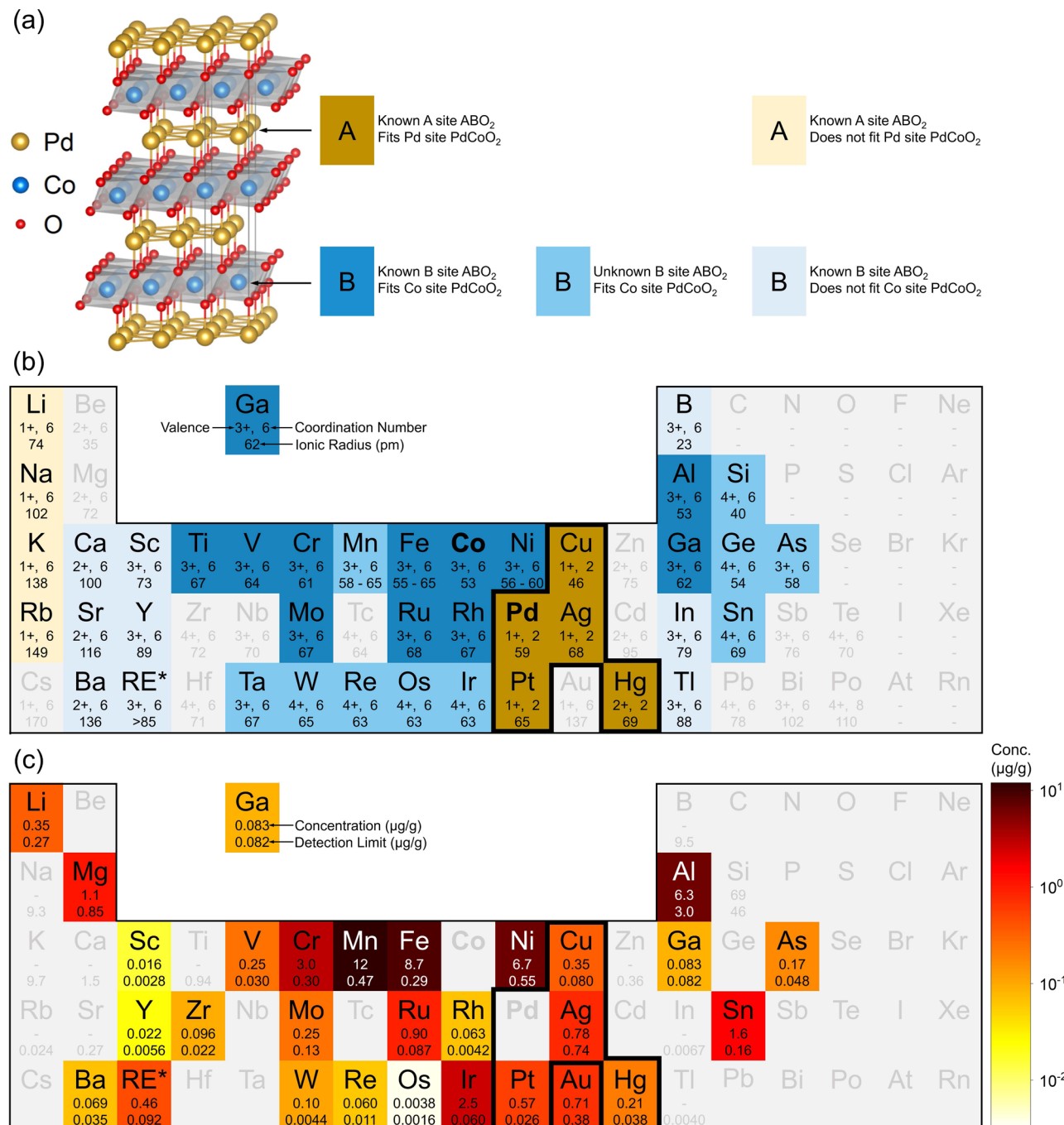

**Fig. 4 | Impurity analysis of CVT-grown PdCoO₂. a** PdCoO₂ crystal structure. **b** Periodic table highlighting elements relevant to the delafossite structure. As in (**a**), light yellow highlights elements that are known to adopt the A-site in ABO₂ compounds but do not fit the Pd site in PdCoO₂. Dark yellow highlights elements that are known to adopt the A-site in delafossites and do fit the Pd site in PdCoO₂; these are highlighted by the bold black border. Analogously, the lightest blue highlights elements that are known to adopt the B site in ABO₂ compounds but do not fit the Co site in PdCoO₂, and the darkest blue highlights elements that are known to adopt the B site in delafossites and do fit the Co site in PdCoO₂; the intermediate blue shade highlights elements for which it is unknown whether they adopt the B site in ABO₂ compounds but they do fit the Co site in PdCoO₂. In this scheme, dark yellow highlights elements likely to substitute for Pd (only four), while the two darker blue colors highlight elements likely to substitute for Co (>20). Also

shown for each element are the valence, coordination number, and ionic radius[80] in the state most closely relevant to delafossite oxides. (Ranges of radii are for low- to high-spin states). Au is not shaded dark yellow in (**b**); there exists some evidence of Au¹⁺ in linear coordination with O²⁻[S1] but it is uncommon. **c** Periodic table labeled with the detected concentration and detection limit for each element probed in mass spectrometry. All values are weight-basis ppm, averaged over multiple runs and CVT crystals, as in Tables 2–4. The color scale (right) is a log₁₀-scale of average impurity concentration in CVT crystals. As in (**b**), the bold black border highlights elements likely to substitute for Pd; their total concentration is only 1.9 ppm. The entire rare-earth element block appears as "RE" in panels (**b**, **c**); in panel (**c**), the values are sum totals for these RE elements. Na, B, and Si, which are common glassware impurities, were excluded from ICP-MS analyses.

**Table 3 | Summary of average A-site impurity concentrations (in weight-basis ppm) in metathesis/flux- and CVT-grown $PdCoO_2$ crystals, as determined by ICP-MS**

| A-site Elemental Impurities in $PdCoO_2$ | | | | |
|---|---|---|---|---|
| | Ag (µg/g) | Hg (µg/g) | Cu (µg/g) | Pt (µg/g) |
| Flux | 1.69 ± 0.01 | BDL | 0.50 ± 0.03 | 2.22 ± 0.02 |
| CVT | 0.78 ± 0.06 | 0.21 ± 0.02 | 0.4 ± 0.2 | 0.6 ± 0.2 |
| DL | 0.74 | 0.038 | 0.080 | 0.026 |

Shown are the concentrations of the only elements (four) that are known to adopt the A-site in $ABO_2$ delafossites and also fit the Pd site in $PdCoO_2$. The averages are over 2–3 runs on 1–3 crystals, and the uncertainties listed are standard deviations. Na, B, and Si, which are common glassware impurities, were excluded from all analyses. The detection limit (DL) is listed for each element; for concentrations below this detection limit, "BDL" is entered.

**Table 4 | Summary of most prevalent B-site impurity average concentrations (in weight-basis ppm) in metathesis/flux- and CVT-grown $PdCoO_2$ crystals, as determined by ICP-MS**

| Most Prevalent B-site Elemental Impurities in $PdCoO_2$ | | | | | | | |
|---|---|---|---|---|---|---|---|
| | Al (µg/g) | Cr (µg/g) | Mn (µg/g) | Fe (µg/g) | Ni (µg/g) | Sn (µg/g) | Ir (µg/g) |
| Flux | 82 ± 2 | 3.22 ± 0.09 | 13.3 ± 0.2 | 9.2 ± 0.2 | 15.0 ± 0.2 | 2.49 ± 0.04 | 1.08 ± 0.01 |
| CVT | 6.3 ± 0.8 | 3 ± 1 | 12 ± 6 | 9 ± 2 | 7 ± 3 | 1.61 ± 0.03 | 3 ± 1 |
| DL | 3.0 | 0.30 | 0.47 | 0.29 | 0.55 | 0.16 | 0.06 |

Shown are the concentrations of the seven most prevalent elements that fit the Co site in $PdCoO_2$. The averages are over 2–3 runs on 1–3 crystals, and the uncertainties listed are standard deviations. Na, B, and Si, which are common glassware impurities, were excluded from all analyses. The detection limit (DL) is listed for each element.

## Discussion

The conclusion from the above is that the prototypical metallic delafossite $PdCoO_2$ is by no means ultrapure in general. Overall metals-basis impurity concentrations decrease from ~250 ppm in metathesis/flux crystals to ~50 ppm in CVT crystals but remain substantial. Crystal-chemical principles, however, dictate that the great majority of substitutional impurities must reside on the Co site, the relatively rare valence and coordination on the Pd site resulting in few elemental impurities (particularly Cu, Ag, and Pt) being capable of substitution. Sublattice purification ratios of ~1% thus arise, limiting the substitutional impurities on the Pd site to ~1 ppm. Our simple conclusion from these findings is that the electronic conduction in metallic delafossites such as $PdCoO_2$, which is strongly restricted to Pd $d$ states at the Fermi level[5,10,18–21], thus occurs in highly pure Pd planes, with minimal substitutional impurities. This sublattice purification naturally explains how ultralow residual $\rho_{ab}$ and extraordinary mean-free-paths can arise in metallic delafossite single crystals, despite the relatively dirty methods employed in their synthesis[5,54]. Directly addressing a question posed in recent work[54], the $BO_6$ octahedral planes in metallic delafossites thus do host substantial disorder, in contrast to the Pd/Pt sheets, which were deduced to be highly perfect[54]. The apparent contradiction of sizable magnetic impurity concentrations in crystals with high RRR is thus naturally resolved, as the majority of substitutional impurities, including magnetic ones, populate only the $CoO_6$ octahedra.

The above picture is predicated, however, on two key assumptions: that the ~1 ppm deduced impurity concentrations in the Pd planes are sufficiently low to facilitate the observed residual $\rho_{ab}$ and mean-free-paths, and that the significant impurity concentrations in the $BO_6$ octahedral planes have sufficiently little impact on conduction in the Pd sheets. The first assumption can be addressed using the 2D unitary scattering approach recently applied by Sunko et al., which was validated in irradiated metallic delafossite crystals[54]. This essentially assumes the strongest possible $s$-wave scattering[54] in a Drude model, resulting in $\rho = \frac{4h}{e^2}\frac{n_d}{n}$, where $\hbar$ and $e$ are the reduced Planck constant and electronic charge, $n_d$ is the areal defect density, and $n$ is the volume carrier density. Based on this, our deduced residual $\rho_{ab}$ of 4 nΩcm would require a 2D defect density of $6 \times 10^9$ cm$^{-2}$ in CVT-grown $PdCoO_2$. This is in striking agreement with our determined Pd-sheet

impurity density, which corresponds to $5 \times 10^9$ cm$^{-2}$ based on the 1.9 ppm of A-site impurities in Table 2. (This calculation takes into account the exact A-site impurity elements in Tables 2 and 3, and their relative masses). This indicates quantitative agreement between our deduced A-site impurity concentration and measured residual resistivity. This extends also to flux/metathesis crystals, where the A-site impurity concentration of 4.4 ppm in Table 2 would be expected to generate a residual $\rho_{ab}$ of 7 nΩcm, very close to measured values[17]. (Again, this calculation takes into account the exact A-site impurity elements in Tables 2 and 3, and their relative masses; the distribution of impurities is different for CVT and metathesis/flux crystals). The second assumption is also supported by the recent work of Sunko et al., which explicitly concluded that disorder in the B-O layers must be efficiently screened;[54] the extent of this disorder was not determined in that paper, but is now known to be substantial (of order 100's of ppm).

Density functional theory (DFT) calculations provide further support for the above. These were performed (see "Methods" section) on $PdCoO_2$ lightly substituted with Pt, Ag, Fe, and Al, i.e., two known impurities on the Pd site and two prevalent impurities on the B site. $3 \times 3 \times 3$ supercells containing a single substituent atom were employed, i.e., 3.7% concentrations, as a computationally tractable approximation to the dilute limit. Unfolding of supercell bands was used to determine the effect of these impurities on the band structure and density-of-states (DOS) within the $PdCoO_2$ primitive cell. Figure 5 shows the resulting band structures in the left panels (a–d) and the corresponding DOS in the right panels (e–h). As might be expected, dilute Pt impurities on the A-site (Fig. 5a, e) have little effect on the band structure and DOS. The Pd $d$ band crossing the Fermi energy ($E_F$) has no visible distortion at $E_F$ in Fig. 5a, and the atom-projected DOS in Fig. 5e is of very similar form for Pd and Pt. Note here that the simulated $Pd_{0.963}Pt_{0.037}CoO_2$ has 26-times more Pd than Pt but no normalization is applied in Fig. 5e–h, simply to promote visibility of the impurity DOS. For Ag impurities (Fig. 5(b, f)), as might also be expected, the impacts on electronic structure are more significant. Some minor distortion of the Pd $d$ band is visible, an impurity band forms ~100 meV below $E_F$, and there is some non-negligible weight crossing the Fermi level. Ag is thus a likely contributor to the unitary-limit scattering discussed above.

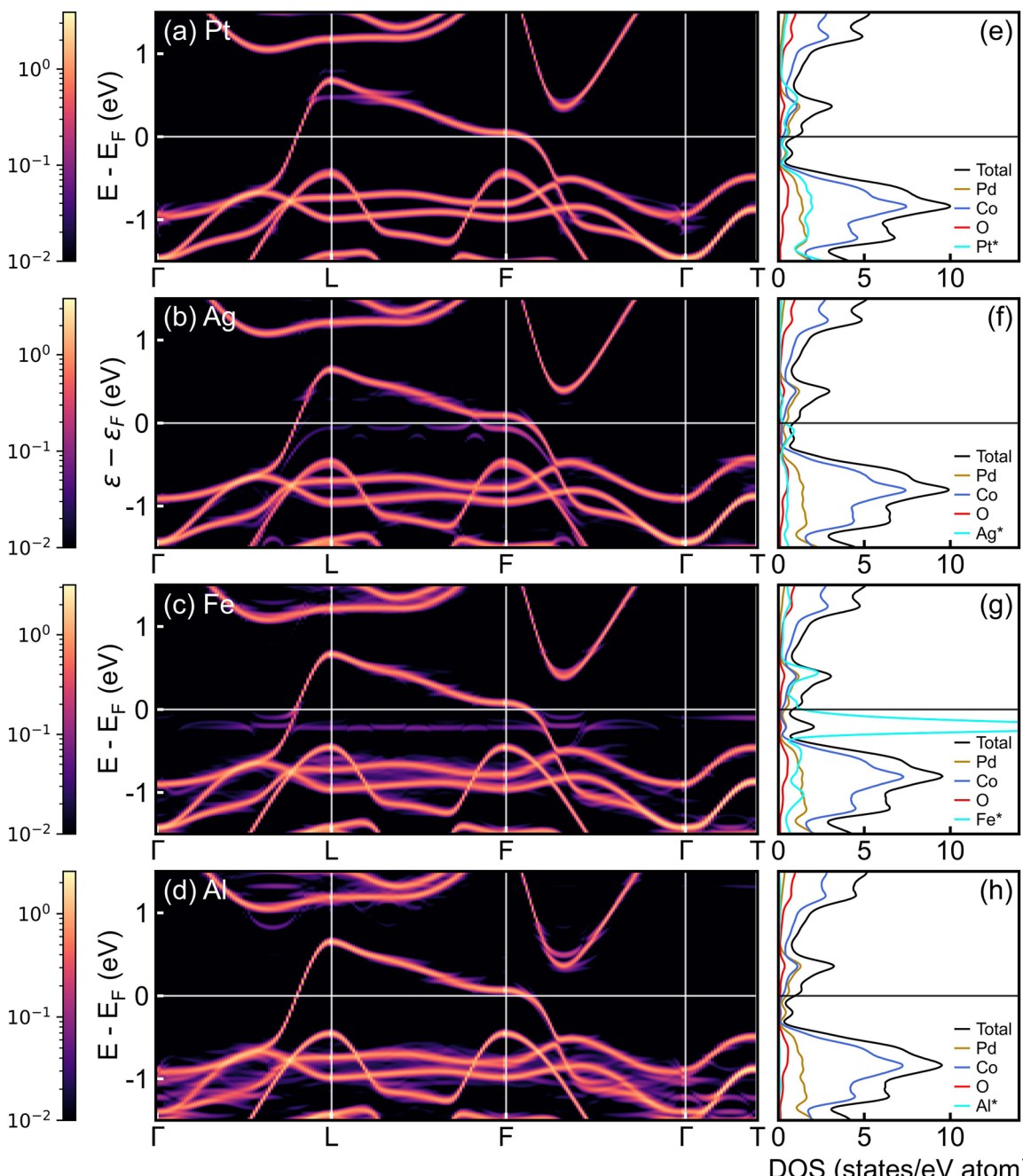

**Fig. 5 | DFT-calculated electronic band structure and density-of-states of lightly-substituted PdCoO₂. a–d** Band structure of $Pd_{1-x}Pt_xCoO_2$, $Pd_{1-x}Ag_xCoO_2$, $PdCo_{1-x}Fe_xO_2$, and $PdCo_{1-x}Al_xO_2$, respectively, for $x = 0.037$, i.e., 3.7% substitutional impurity density. Shown is the high-symmetry $k$-path Γ-L-F-Γ-T, in the first Brillouin zone, with energies relative to the Fermi energy, $E_F$. The intensities are on a $\log_{10}$ scale depicting the weight of the unfolded primitive cell bands obtained from the supercell bands *via* the unfolding method[77,78]. **e–h** Corresponding density-of-states

(DOS). Shown are the total DOS (black), and the atom-projected DOS of Pd (dark yellow), Co (blue), O (red), and the respective impurity (cyan). The DOS values are for the primitive cell, calculated by dividing by the corresponding number of atoms in the supercell. The atom-projected DOS curves for the impurity atoms (labeled with an asterisk) should thus be scaled by a factor of 0.037 for direct comparison; they are plotted here unscaled for improved visibility.

As might also be anticipated, the situation for Fe impurities on the B site is more complex. A narrow impurity band forms below $E_F$ (Fig. 5c), generating a clear peak in the Fe-projected DOS in Fig. 5g, much larger than in Fig. 5f. Nevertheless, this main impurity band peak is centered at ~0.2 eV below $E_F$ (many times $k_BT$ at the temperatures of interest in this work), and is quite narrow, meaning that the impact at $E_F$ remains negligible even for Fe impurities. This directly supports the above conclusion that typical transition-metal B-site impurities (which are present at much lower concentrations in experiment) have only weak impact on conduction in the Pd planes of PdCoO₂. For Al

impurities (Fig. 5d, h), the absence of $d$ electrons results in no significant effect on the band structure or DOS in the energy range in Fig. 5d, h, the only impact being some smearing of existing bands. This smearing is likely related to the larger structural distortions induced by Al impurities relative to Fe and Pt. The Pd-Pd length for Pd ions bonded via O in $AlO_6$ octahedra shifts by as much as 0.6%, for example, 1.5 times more than the equivalent for Fe impurities, and 5 times more than for Pt. Similarly, the average bond length in $AlO_6$ octahedra differ by ~0.5% from the $CoO_6$ octahedra in pristine PdCoO₂, much more than for the Fe case (0.02%). Figure 5 thus establishes that the B-site

impurities Fe and Al indeed have greater impact on the electronic structure than an A-site impurity such as Pt, but even Fe and Al have little meaningful impact in the vicinity of $E_F$, supporting our arguments.

It should be emphasized that the above analysis focuses on substitutional impurities. While interstitial impurities should also be considered, the delafossite structure is unlikely to host interstitial sites in which the ions in Fig. 4b can realistically be accommodated. The most obvious interstitial space involves an extra ion bonded to two $O^{2-}$ ions in the adjacent B-O layer, for which DFT calculations suggest a formation energy as high as 6.3 eV in the case of Pd interstitials[54], consistent with the low interstitial volume. The equilibrium concentrations of such defects would thus be negligible, and would decrease for larger Ag, Pt, and Hg interstitials. With respect to vacancies, the above arguments strongly support that only Pd vacancies are of any real potential relevance for *a-b* plane metallic transport, as Co and O vacancies obviously reside in the insulating Co-O octahedral layers. The above quantitative agreement between measured residual resistivities and deduced Pd-layer substitutional impurity densities then strongly argues that any additional Pd vacancy effects are minor, in agreement with Sunko et al. [54]. Beyond point defects, line and areal defects should be considered. As alluded to in the Introduction, however, current indications are that the densities of such defects are extraordinarily low in single-crystal metallic delafossites. This is supported by TEM observations[54,60], deductions from irradiation-induced defect studies[54], the substantial formation energies for most point defects in $PdCoO_2$[54], and the various additional observations of structural quality in the current paper, most notably the X-ray rocking curve widths (Fig. 2c inset). Substitutional impurities thus appear to be the key defects in metallic delafossites, which have been comprehensively elucidated here. Future work to more completely understand the reasons for the low density of line and areal defects in these compounds is nevertheless clearly important.

In summary, this work establishes CVT as a promising approach to bulk-single crystal growth of $PdCoO_2$, with potential applicability to other metallic delafossites such as $PdCrO_2$ and $PdRhO_2$. Compared to standard metathesis/flux methods, CVT generates orders-of-magnitude gains in crystal size and mass, along with the highest structural quality, highest purity, lowest magnetic impurity concentration, and highest RRR yet reported. Nevertheless, detailed trace impurity analysis of $PdCoO_2$ single crystals by ICP-MS reveals that, contrary to the naïve view, these metallic delafossites are not generally ultrapure; standard metathesis/flux methods result in ~250 ppm overall impurity concentrations, falling to ~50 ppm in CVT crystals. Simple crystal-chemistry-based analyses show that the vast majority of these impurities are forced to reside in the Co-O octahedral planes, however, only a small fraction being capable of substituting for Pd (typically a total of ~1 ppm of Cu, Ag, and Pt). The sublattice purification ratio is thus ~1%, resulting in ultrahigh purity in the Pd planes in which electronic conduction takes place, largely unaffected by the impurities in the $CoO_6$ octahedra. This sublattice purification mechanism is therefore central to the ultrahigh low-$T$ conductivity of metallic delafossites, resolving the apparent contradiction that they appear impure from magnetometry yet highly pure in electronic transport. These results significantly demystify the outstanding low-$T$ conductivity and mean-free-path in metallic delafossites, setting the stage for further advances with this extraordinary materials class. As an example, the analyses in Refs. 28,33 place $PdCoO_2$ in a situation where the momentum-conserving scattering rate is larger, but not much larger, than the momentum-relaxing scattering rate, leaving the system in a crossover regime between ballistic and viscous electronic transport. The momentum-conserving scattering is unlikely to be electron-electron scattering at the temperatures of interest[33] but could be electron-impurity scattering. It would thus be interesting to

understand whether the Co-O plane impurities elaborated here could be responsible for such. More broadly, the concepts developed in this work can also be expanded to other metallic and semiconducting delafossites, and potentially to other complex crystal families with very different valence/chemical environments for different cations.

## Methods

### Crystal growth
$PdCoO_2$ crystals were grown by two methods: the established metathesis/flux process[6] and our CVT process[18]. The established process utilizes the metathesis reaction $PdCl_2 + Pd + Co_3O_4 \rightarrow 2PdCoO_2 + CoCl_2$[6]. Powders of $PdCl_2$ (ThermoFisher Scientific, 99.999% purity), Pd (Sigma-Aldrich, 99.995%), and $Co_3O_4$ (Alfa Aesar, 99.9985%) were mixed in a 1.5:1:1 molar ratio, then loaded and vacuum sealed (at $10^{-6}$ Torr) in quartz ampoules; excess $PdCl_2$ is reported to promote crystallization of $PdCoO_2$[14]. Ampoules were then heated to 700–750 °C at a ramp rate of 300 °C/h in a box furnace, dwelled for 40–150 h, cooled to 400 °C at 40–60 °C/h, then furnace cooled to ambient. These conditions were fine-tuned to optimize yield and phase purity. To remove excess $PdCl_2$ and the reaction by-product $CoCl_2$, the crystalline $PdCoO_2$ products (shown in Fig. 1b) were soaked in ethanol overnight then repeatedly washed in boiling ethanol until the effluent ran clear. Minor unreacted Pd and $Co_3O_4$ phases were often detected in these products by PXRD (see Supplementary Note 2). The largest crystals found in the products were $0.8 \times 0.6$ mm$^2$ laterally and 0.017 mm thick (Fig. 1b, inset).

For CVT, ~1 g of coarse-grained $PdCoO_2$ product from the above metathesis reaction was mixed with ~120 mg of $PdCl_2$ powder, then loaded and vacuum sealed (at $10^{-6}$ Torr) in quartz ampoules. Ampoules were then placed in a two-zone tube furnace with the reagents at one end of the ampoule (Fig. 1a). CVT was performed at a range of temperatures, spanning 740–800 °C for the hot zone and 680–750 °C for the cold-zone. Hot- and cold-zone temperatures of 760 and 710 °C were found optimal in terms of crystal size and complete transport (i.e., yield). For the first 3 days, the empty end of the tubes was maintained as the hot end, to clean the growth zone. The temperature gradient was then inverted for 13 days to establish the reagent-filled end of the tubes as the hot zone and the originally empty end as the cold/growth zone (Fig. 1a). We believe CVT to proceed via decomposition of $PdCl_2$ (above ~600 °C[62,63]), followed by the reaction $PdCoO_2(s) + 2Cl_2(g) \leftrightarrow PdCl_2(g) + CoCl_2(g) + O_2(g)$ (Fig. 1a). Multi-crystals up to $12 \times 12$ mm$^2$ laterally and 0.3 mm thick result from this process (Fig. 1c). Single crystals were then isolated by gently breaking the multicrystals apart, generating up to $6.0 \times 4.0$ mm$^2$ lateral sizes, with thickness up to 0.17 mm (see Fig. 1c, inset, for an example crystal). Excess chlorides were then removed by washing $PdCoO_2$ crystals in boiling ethanol. We are aware of no prior report of such CVT growth of metallic delafossites, although one approach to $PdRhO_2$ growth did employ a CVT-like temperature gradient[56] but with clear qualitative differences with our methods and findings.

### Structural and chemical characterization
Optical microscope images of crystals were acquired with a Zeiss Axiovert A1 MAT inverted microscope, while SEM was performed in a JEOL JSM-6010PLUS/LA. EDS data were taken with an EDS detector integrated in the SEM, at 20 kV accelerating voltage. PIXE spectra were acquired in a MAS 1700 pelletron tandem ion accelerator (National Electrostatic Corp.) at 50 μC dose from a 4 MeV $He^{2+}$ beam; PIXE data were analyzed using GUPIXWIN[70]. Laue diffraction, PXRD, and single-crystal HRXRD (including rocking curves) were taken using Photonic Science back-reflection Laue, Rigaku MiniFlex 600, and Rigaku SmartLab XE diffractometer systems, respectively, the latter two with CuK$_\alpha$ radiation. PXRD analysis employed JADE for whole pattern fits, which were used to determine *a* and *c* lattice parameters. Laue analysis employed Lauesim[72], modified for hexagonal symmetry.

## Electronic transport and magnetometry measurements

Single crystals for electronic transport were first cleaved into bar shapes (Fig. 3a, inset) and affixed with GE varnish to $Al_2O_3$ wafers. Al contact wires were attached in a four-point in-line geometry to the top surface of the $PdCoO_2$ using ultrasonic wire bonding. $T$-dependent (~4–300 K) resistance measurements were then performed in a Janis cryostat/superconducting magnet, using a Lakeshore 370 AC resistance bridge and Lakeshore 3708 preamplifier/channel scanner (at 13.7 Hz and 10 mA). As alluded to in the main text and described in detail in Supplementary Note 3, great care was taken to accurately determine and account for instrumental voltage offsets, which are important due to the very low low-$T$ resistance of $PdCoO_2$. A specialized protocol for this was developed and tested on zero-resistance superconducting V films. As also described in Supplementary Note 3, finite-element COMSOL simulations were performed to support the interpretation of the transport results, particularly with respect to the issue of systematic errors associated with the $\rho_c/\rho_{ab}$ anisotropy. To minimize contamination for magnetometry measurements, crystals were first ground with thoroughly cleaned nonmagnetic tools then washed in ethanol to remove $CoCl_2$. The latter could occur not only at the original crystal surfaces but also at internal inclusions/voids[16]. Magnetometry was then done in a Quantum Design Physical Property Measurement System equipped with vibrating sample magnetometry (VSM) and high-$T$ options, from 2 to 600 K in magnetic fields to 7 T.

## Impurity analysis

For trace impurity analyses, ethanol-washed crystals were microwave digested in aqua regia (a 1:3 mixture of 70%-concentrated trace-metal-grade nitric acid (Ricca) and 36%-concentrated trace-metal-grade hydrochloric acid (Fisher Scientific)), or concentrated nitric acid, in sealed Teflon pressure vessels. A CEM MARS 5 microwave digestion system was employed, reaching 200 °C and ~15 bar for 2 h. Solutions were allowed to slow cool to room temperature overnight before serial dilution and analysis; no visible solid remained. Trace element analysis in a Thermo Scientific iCap TQ triple quadrupole ICP-MS system was performed by standard addition methods using SPEX CertiPrep and Thermo-Fisher Scientific multielement standards. Internal standard sets included: Sc, Y, Ir for alkali metals, alkaline-earth metals, non-precious transition metals, metalloids, and heavy metals; Sc, Y, Bi for precious metals; V, Rb, Cs, Bi for Sc, Y, and rare-earth elements. ICP-MS analytical details for each element are listed in Supplementary Table 6.

## DFT calculations

First-principles density functional theory calculations were performed with the Quantum ESPRESSO package using norm-conserving pseudopotentials, with the exchange-correlation energy approximated by the Perdew-Burke-Ernzerhof functional[73–75]. The energy cutoff for the plane waves was set to 100 Ry. Dilute impurities on the Pd- and Co-sites (i.e., $Pd_{1-x}Z_xCoO_2$ and $PdCo_{1-x}Z_xO_2$) were simulated in a $3 \times 3 \times 3$ supercell with 108 atoms, with one Pd or Co atom replaced, corresponding to $x = 0.037$. Pt and Ag substituted for Pd, and Fe and Al substituted for Co were considered. Both the unit cell shape and atomic positions were relaxed for electronic structure calculations. The band structure and DOS were calculated on a Γ-centered Monkhorst-Pack grid of $4 \times 4 \times 4$ $k$ points in the Brillouin zone of the supercell. The unfolded band structures in Fig. 5 were obtained using the bandUPpy code[76–78]. The unfolding method projects a wave vector **K** in the supercell Brillouin zone onto a corresponding wave vector **k** in the primitive cell Brillouin zone via a supercell reciprocal lattice vector. This unfolding evaluates the spectral weight or probability of finding a set of primitive-cell Bloch states at wave vector **k** that are contributing to the supercell eigenstates at wave vector **K** at the same energy. The specific method adopted here[77] finds the effective band structure for doped $PdCoO_2$ by using the spectral function to calculate the weight, which is given by the number of primitive cell bands that are crossing within a small energy window around the band center.

## Data availability

All data needed to evaluate the conclusions presented in this study have been deposited in DRUM (Data Repository for the University of Minnesota) at: https://doi.org/10.13020/0k3v-jj67. Any additional digital data are available from the corresponding author upon request.

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

## Acknowledgements

We thank Prof. Xinyuan Zhang for useful advice on ICP-MS analysis. This work was primarily supported by the US Department of Energy through the University of Minnesota (UMN) Center for Quantum Materials, under Grant No. DE-SC0016371 (CL). Parts of this work were carried out in the Characterization Facility, UMN, which receives partial support from the National Science Foundation through the MRSEC (Award Number DMR-2011401) and NNCI (Award Number ECCS-2025124) programs. The Minnesota Supercomputing Institute at UMN provided resources that contributed to the research reported within this paper.

## Author contributions

CL conceived the study and supervised its execution. YZ and FT grew the crystals, which were structurally and chemically characterized by YZ, FT, GH, MM, and JG-B. YZ, FT, BK, JR, and VC performed the transport and magnetometry measurements. YZ, SB, and YT performed Laue characterization and analysis. YZ, GE, and WS performed mass spectrometry measurements and analysis. PS, TB, and RF performed the theoretical work. CL and YZ wrote the paper, with input from all authors.

## Competing interests

The authors declare no competing interests.
