## [Peer Review File · Nature Communications]

Crystal-Chemical Origins of the Ultrahigh Conductivity of Metallic DelafossitesREVIEWER COMMENTS

Reviewer #1 (Remarks to the Author):

In this manuscript, Zhang et al. report on the CVT growth of millimeter-sized crystals of PdCoO₂ that show a record residual resistivity ratio. Based on thorough impurity analysis, they propose that most impurities in PdCoO₂ substitute B-site ions, leaving the conductive Pd sheets highly pure. I found the discussion interesting and agree that the reported technique of growing large crystals accelerates research of metallic delafossites. I think this is an important contribution and should be published in some form. I list some comments that could improve the manuscript.

1. Throughout the discussion, the authors focus on impurities that substitute A or B sites in ABO₂. Other types of defects can contribute to the residual resistivity, e.g., cation and oxygen vacancies. The authors should include a discussion on possible defects other than impurities to make their argument comprehensive.

2. In Fig. 5, the authors argue that Pt impurities have little effect on the electronic structure and DOS. How about Ag⁺ impurities? The Ag⁺ are contained in the largest concentration as in Fig. 5(c), and have one extra valence electron compared with Pd⁺ that I expect to influence on the band structure?

3 On p. 3, "... results in a strikingly simple electronic structure where a single Pd d band crosses the Fermi level, ...". The conduction band of PdCoO₂ can also have a contribution from Pd 5s states. [Physica B 245, 157 (1998)]

4 In p. 4, "... potential applications are being explored, ...". The authors may include two more important applications: transparent conductors [APL Mater. 6, 046107 (2018)] and terahertz sources [Adv. Mater. 35, 2305622 (2023)].

5 In p. 12, "our measured R of ~440 is a record in PdCoO₂, which can be compared to 376 in prior metathesis/flux-grown crystals ...". The author should compare their result with the largest value reported: RRR ~ 407 by Takatsu et al. [J. Phys. Soc. Jpn. 76, 104701 (2007)].

Reviewer #2 (Remarks to the Author):

Referee report on 'Crystal-Chemical Origins of the Ultrahigh Conductivity of Metallic Delafossites' by Yi Zhang et al

In this paper, Zhang et al present a thorough and beautiful study of the materials science behind the incredibly high conductivities seen in the metallic delafossites. They concentrate on PdCoO₂ but there is little doubt that their conclusions would apply to PtCoO₂, PdCrO₂ and PdRhO₂ as well. By adopting chemical vapour transport as a growth technique for PdCoO₂, they succeed in growing much bigger crystals than had been possible previously, giving them access to highly sensitive chemical analysis techniques, notably inductively coupled plasma mass spectrometry. These analyses confirm what had been suspected, but not proven, previously: the perovskite layers of the materials can host standard levels of impurities, consistent with expectation for the relatively unrefined growth techniques that are used in the field. In contrast, the conducting Pd layers are ultrapure, in quantitative agreement with conclusions drawn previously in the paper cited as Ref. 49 in their manuscript.

This paper is a shining example, too rare in the modern world, of the way science should be done. The work is careful and of high quality, and clearly explained in the manuscript. The findings are important both in terms of materials science and because they will drive progress in an important field of research. I strongly recommend publication in Nature Communications.

I do have a few comments for the authors though.

1. Personally I think that the conclusion about purity could be worded slightly differently. Rather than saying 'the delafossites are not ultrapure' it might be better to stress even more strongly that while they are not *overall* ultrapure, the conducting planes are.

The context of suggesting this is that there are now quite a number of topological materials (and others) in which the wavefunction structure of the conducting bands leads to strongly suppressed backscattering. They can be highly impure on all their crystalline sites, but still have high conductivity. In the delafossites the analogue would be to have really impure conductive planes but still get high conductivity. By stressing that the crystal chemistry of the delafossites allows for ultrapure planes, one can distinguish them from the 'strongly suppressed backscattering' scenario. I think it is useful to do this, because it is in some senses more remarkable (and rarer) to have ultrapure planes than it is to have strong backscattering suppression.

This point is made very well in the body of the paper, but the statement 'not ultrapure' might be adopted as a catch phrase and I think that is better avoided.

2. I think that the wording 'previously unconsidered' when describing sublattice purification in the abstract is not completely accurate. As the authors stress in the manuscript, the idea that the conducting planes are purer than the perovskite layers has been considered by a number of other authors. That does not take away from the current work – it is one thing to consider it but another thing entirely to prove it. Nevertheless, I think the authors should consider a rewording.

3. On line 475 the authors state that 'other point, line, and areal defects should be considered'. This is correct, but it might be a good idea to mention perhaps the most famous of these, namely vacancies. I know that the current analysis procedures have no sensitivity to vacancies, but some quick discussion might be nice: the fact that the in-plane impurity concentration gives a quantitative match to the resistivity suggests that the vacancy concentration in the planes is incredibly low.

Reviewer #3 (Remarks to the Author):

I found this to be a strong paper on a topic of considerable current interest. The detail on crystal growth and characterization is admirably thorough. The particular advance in knowledge here is a resolution of the reason for the exceptionally high carrier mobility in this material, a property that has been a puzzle for some time. I think their conclusion was nearly inevitable, and suspected by other researchers, but it is good to see such a solid job of coming to a definitive answer.

I recommend publication of the article, once the authors have considered my comments below.

1. The comparison of lattice constants (both the means and standard deviations) for crystals grown by the two methods is not statistically very sound. Just by inspection, it seems true that the standard deviation is smaller for the CVT-grown crystals, but they should at the very least talk about sample size when making such a comparison. The statement on the difference between the means is not statistically supported. Even at the 1-sigma level, the distributions overlap considerably. At the very least they would need to discuss uncertainty of the mean, better still would be a proper statistical test. With the right checks for validity, this is most likely a t-test in this situation. Our field tends to be unsophisticated in its statistical treatments and this is a situation where the authors have everything they need to do a more statistically robust comparison.

2. I find the poor measurement geometry of the DC resistivity (top leads) disappointing for such strong work otherwise. I am reluctant to say go back and do the measurement better. But if the authors don't do a much improved measurement, I would suggest a very soft claim about having improved over the current state of the art. See my next point in relation to this.

3. The statements that these are better crystals than those grown via flux methods is likely true, but the improvement is not all that dramatic, and the argument has the weaknesses mentioned above. That does not change the more important conclusion of the paper, which is the "self-purifying" nature of this system. To some extent the claims of sample improvement obscure this more important conclusion.

Reviewer #4 (Remarks to the Author):

The manuscript "Crystal-Chemical Origins of the Ultrahigh Conductivity of Metallic Delafossites" by Zhang et al., is a good work which shows a growth methodology to achieve very high quality oxides of the family of delafossites. The latter have attracted attention because of many properties, in particular, for their high conductivity which is even higher than noble metals. Here, the authors argue the chemical origin behind the transport.

The paper is well written and nice to follow. I found however, many statements which are oversold and many other which I find myself in disagreement. The authors immediately state that the origin of the high conductivity is poorly understood. I disagree, there are several works, both theoretical and experimental, i.e., DFT, DMFT, Transport, quantum oscillations, ARPES, transport under irradiation etc., which have found quite solidly that the high conductivity of these systems is due to the s-dz orbitals hybridisation. This was, in agreement with the author conclusions attributed to Pd mobilities.

On the one hand, the growth is genuinely impressive and the crystals obtained are, as the authors fairly report, much larger in size than the current ones. This is an excellent achievement. However, I do feel that the advance, compared to current existing and published literature is not enough to guarantee the publication in Nature Communications. I think the paper, which is a fine work, can find a more suitable fit in a more material oriented journal. At the moment, I do not see the advance to justify the fit with the current journal.

**Response to Review Comments, NCOMMS-23-42160,
“Crystal-Chemical Origins of the Ultrahigh Conductivity of Metallic Delafossites”,
Zhang *et al.***

We would first like to thank the reviewers for their thoughtful and detailed comments on our work, which we are certain have resulted in a yet stronger version of the manuscript. Below, we provide point-by-point responses to these comments, along with brief summaries of the ensuing manuscript changes in red. In the attached manuscript, the changes are also highlighted in red.

Reviewer 1

In this manuscript, Zhang *et al.* report on the CVT growth of millimeter-sized crystals of PdCoO₂ that show a record residual resistivity ratio. Based on thorough impurity analysis, they propose that most impurities in PdCoO₂ substitute B-site ions, leaving the conductive Pd sheets highly pure. I found the discussion interesting and agree that the reported technique of growing large crystals accelerates research of metallic delafossites. I think this is an important contribution and should be published in some form. I list some comments that could improve the manuscript.

We thank the reviewer for the acknowledgment of the importance of our work, the opinion that it will accelerate research on metallic delafossites, and the recommendation that it should be published.

1. Throughout the discussion, the authors focus on impurities that substitute A or B sites in ABO₂. Other types of defects can contribute to the residual resistivity, e.g., cation and oxygen vacancies. The authors should include a discussion on possible defects other than impurities to make their argument comprehensive.

This is an important point, we agree. To more fully address it, **we have now expanded the discussion of defects other than impurities, which commences on page 22**. In particular, a discussion regarding vacancies has been added to this existing section. The key points made in this discussion are two-fold. First, as our work and the recent work of others shows, conduction in these compounds is almost entirely dominated by the Pd planes, with very little effect from the CoO₆ octahedral layers. This means that it is Pd vacancies, rather than Co or O vacancies, that are most important. (This is far from the usual situation in complex oxides). The second point, nicely stated by Reviewer 2, is that the quantitative agreement between our measured residual resistivities and our measured impurity concentrations directly implies that substitutional impurities are entirely dominant in the defect scattering in these compounds, *i.e.*, that the native Pd vacancy concentration must be relatively negligible. This is additionally supported by Ref. 52.

2. In Fig. 5, the authors argue that Pt impurities have little effect on the electronic structure and DOS. How about Ag⁺ impurities? The Ag⁺ are contained in the largest concentration as in Fig. 5(c), and have one extra valence electron compared with Pd⁺ that I expect to influence on the band structure?

Thank you for urging us to look into this specific point. To address it, we calculated the electronic structure for Ag impurities in PdCoO₂ also, using the same methods as for the other impurities we

considered. Fig. 5 has thus been modified to add new panels for the Ag case, along with a short associated discussion. The essence of the latter is that Ag impurities, as one would expect, have larger impact on the electronic structure than Pt impurities. While the primary Pd *d* band is largely unaffected by Ag substitution, unlike Pt impurities, Ag impurities generate a distinct impurity band. This is centered about 100 meV below the Fermi level but generates some small weight at the Fermi level, indicating that non-negligible effects on transport may well occur. This is consistent with our interpretation that Ag impurities contribute to the unitary-limit scattering described in the Discussion section of our manuscript.

3 On p. 3, "... results in a strikingly simple electronic structure where a single Pd *d* band crosses the Fermi level, ...". The conduction band of PdCoO₂ can also have a contribution from Pd *s* states. [Physica B 245, 157 (1998)].

Stimulated by this comment, we have modified the manuscript to explicitly point out that such a contribution from Pd *s* states has been reported. We wish to point out, however, that our own DFT calculations do not support this conclusion. As shown in Fig. R1 below, for example, the DOS at all energies close to the Fermi level is almost devoid of contributions from Pd *s*. Instead, as remarked in our prior work (Ref. 18), there are non-negligible contributions from Co near the Fermi level.

Figure R1. Energy dependence of the DFT-calculated DOS of PdCoO₂ based on the calculations reported in this work. The total, Pd *d*, and Pd *s* contributions are shown. Note the negligible contribution from Pd *s* states in this energy range.

4 In p. 4, "... potential applications are being explored, ...". The authors may include two more important applications: transparent conductors [APL Mater. 6, 046107 (2018)] and terahertz sources [Adv. Mater. 35, 2305622 (2023)].

These are good points. These applications are now mentioned and these citations have been added.

5 In p. 12, "our measured R of ~440 is a record in PdCoO₂, which can be compared to 376 in prior metathesis/flux-grown crystals ...". The author should compare their result with the largest value reported: RRR ~ 407 by Takatsu et al. [J. Phys. Soc. Jpn. 76, 104701 (2007)].

Our comparison point for RRR values was chosen based on the use of the more accurate FIB-patterned geometries for transport measurements. **We have revised the manuscript to make clear that we primarily compare our measured values to 376 for this reason.** The above reference is nevertheless cited.

Reviewer 2

In this paper, Zhang et al present a thorough and beautiful study of the materials science behind the incredibly high conductivities seen in the metallic delafossites. They concentrate on PdCoO₂ but there is little doubt that their conclusions would apply to PtCoO₂, PdCrO₂ and PdRhO₂ as well. By adopting chemical vapour transport as a growth technique for PdCoO₂, they succeed in growing much bigger crystals than had been possible previously, giving them access to highly sensitive chemical analysis techniques, notably inductively coupled plasma mass spectrometry. These analyses confirm what had been suspected, but not proven, previously: the perovskite layers of the materials can host standard levels of impurities, consistent with expectation for the relatively unrefined growth techniques that are used in the field. In contrast, the conducting Pd layers are ultrapure, in quantitative agreement with conclusions drawn previously in the paper cited as Ref. 49 in their manuscript.

This paper is a shining example, too rare in the modern world, of the way science should be done. The work is careful and of high quality, and clearly explained in the manuscript. The findings are important both in terms of materials science and because they will drive progress in an important field of research. I strongly recommend publication in Nature Communications.

We are delighted by the reviewer's assessment of our work.

I do have a few comments for the authors though.

1. Personally I think that the conclusion about purity could be worded slightly differently. Rather than saying 'the delafossites are not ultrapure' it might be better to stress even more strongly that while they are not *overall* ultrapure, the conducting planes are.

The context of suggesting this is that there are now quite a number of topological materials (and others) in which the wavefunction structure of the conducting bands leads to strongly suppressed backscattering. They can be highly impure on all their crystalline sites, but still have high conductivity. In the delafossites the analogue would be to have really impure conductive planes but still get high conductivity. By stressing that the crystal chemistry of the delafossites allows for ultrapure planes, one can distinguish them from the 'strongly suppressed backscattering' scenario. I think it is useful to do this, because it is in some senses more remarkable (and rarer) to have ultrapure planes than it is to have strong backscattering suppression.

This point is made very well in the body of the paper, but the statement 'not ultrapure' might be adopted as a catch phrase and I think that is better avoided.

We completely understand the reviewer's point. We have addressed this by rephrasing the relevant

sentences in the abstract and conclusion sections. We now make more explicit that the overall purity is not exceptional, but the Pd-plane purity is.

2. I think that the wording 'previously unconsidered' when describing sublattice purification in the abstract is not completely accurate. As the authors stress in the manuscript, the idea that the conducting planes are purer than the perovskite layers has been considered by a number of other authors. That does not take away from the current work - it is one thing to consider it but another thing entirely to prove it. Nevertheless, I think the authors should consider a rewording.

Also understood. To address this, we have simply removed this phrase from the abstract, leaving just the statement that "a "sublattice purification" mechanism is essential to the ultrahigh low-temperature conductivity and mean-free-path of metallic delafossites".

3. On line 475 the authors state that 'other point, line, and areal defects should be considered'. This is correct, but it might be a good idea to mention perhaps the most famous of these, namely vacancies. I know that the current analysis procedures have no sensitivity to vacancies, but some quick discussion might be nice: the fact that the in-plane impurity concentration gives a quantitative match to the resistivity suggests that the vacancy concentration in the planes is incredibly low.

Agreed. This is very similar to point 1 of Reviewer 1. Please see the above response; we have added the requested discussion, along the exact lines the reviewer suggests.

Reviewer 3

I found this to be a strong paper on a topic of considerable current interest. The detail on crystal growth and characterization is admirably thorough. The particular advance in knowledge here is a resolution of the reason for the exceptionally high carrier mobility in this material, a property that has been a puzzle for some time. I think their conclusion was nearly inevitable, and suspected by other researchers, but it is good to see such a solid job of coming to a definitive answer.

I recommend publication of the article, once the authors have considered my comments below.

We thank the reviewer for the positive comments, particularly regarding the thorough and definitive nature of our work, and for the recommendation to publish.

1. The comparison of lattice constants (both the means and standard deviations) for crystals grown by the two methods is not statistically very sound. Just by inspection, it seems true that the standard deviation is smaller for the CVT-grown crystals, but they should at the very least talk about sample size when making such a comparison. The statement on the difference between the means is not statistically supported. Even at the 1-sigma level, the distributions overlap considerably. At the very least they would need to discuss uncertainty of the mean, better still would be a proper statistical test. With the right checks for validity, this is most likely a t-test in this situation. Our field tends to be unsophisticated in its statistical treatments and this is a

situation where the authors have everything they need to do a more statistically robust comparison.

We agree with this point. In hindsight, we should have performed deeper statistical analysis. To this end, we have now executed a t -test on these data. As PXRD is a frequent characterization for us, we had 26 distinct data sets available on metathesis/flux crystals and 18 distinct data sets on CVT crystals. For the a -axis lattice parameter, the t -test supports our claim that the two sets of crystals are different to a p value of ~ 0.0001 . For the c -axis lattice parameter, the equivalent is even smaller. Both values are far beneath often-used thresholds such as 0.05 or 0.01, indicating high statistical confidence in our claim. **We have modified the manuscript to make clear what our sample sizes are, and to provide the above quantitative support for our argument.**

2. I find the poor measurement geometry of the DC resistivity (top leads) disappointing for such strong work otherwise. I am reluctant to say go back and do the measurement better. But if the authors don't do a much improved measurement, I would suggest a very soft claim about having improved over the current state of the art. See my next point in relation to this.

We agree that FIB-patterned samples are undoubtedly more accurate for these types of highly-conductive, anisotropic crystals. However, we make the following points in response:

- As the reviewer is clearly aware, there is a significant learning curve to the FIB patterning of such samples. Ironically, it is even more demanding in our case, as CVT crystals are so much thicker than metathesis/flux on average, making FIBing through the thickness more difficult and time-consuming (unusually thin crystals have to be selected and isolated).
- While our measurements use top contacts, they have been executed as carefully as possible in all other respects, including *via* noise minimization, proper accounting for offsets, and the simulations needed to assess the level of underestimation of the RRR due to the temperature-dependent anisotropy.
- Perhaps most critically, the abovementioned simulations and analysis strongly support that our measured RRR of ~ 440 *must be an underestimate*.

On the basis of these points, we remain confident that the RRR of our crystals is an improvement over prior works. Explicitly, we only claim ~ 440 in this regard, but the true value must be substantially higher. As elaborated further in response to the next point, we feel it is important to also emphasize that RRR is *but one measure* by which our CVT crystals are improved over prior work.

3. The statements that these are better crystals than those grown via flux methods is likely true, but the improvement is not all that dramatic, and the argument has the weaknesses mentioned above. That does not change the more important conclusion of the paper, which is the "self-purifying" nature of this system. To some extent the claims of sample improvement obscure this more important conclusion.

We understand the reviewer's point, but we would like to emphasize that the evidence that these CVT crystals are an improvement over prior work is not based solely on RRR. In fact, the quantitative evidence includes:

- The greatly increased size, amounting to a mass improvement of 5300 for our multicrystals, and 500 for our single crystals.

- The decreased cell volume, which, thanks to the reviewer's above comment, is now made more quantitative.
- The FWHM of the wide-angle X-ray rocking curve of only 0.0089° , the lowest ever reported for a metallic delafossite.
- The lower overall impurity concentration, by a factor of 5, and the lower A-site impurity concentration, by a factor of over 2.
- The higher RRR. Again, we claim only ~ 440 compared to the prior (FIB-based) report of 376, but this is a lower bound on the true result.

If the reviewer thinks it would enhance the paper, we could summarize the above in one specific location in the manuscript? All of this information is currently presented but not in one summary.

Reviewer 4

The manuscript "Crystal-Chemical Origins of the Ultrahigh Conductivity of Metallic Delafossites" by Zhang et al., is a good work which shows a growth methodology to achieve very high quality oxides of the family of delafossites. The latter have attracted attention because of many properties, in particular, for their high conductivity which is even higher than noble metals. Here, the authors argue the chemical origin behind the transport. The paper is well written and nice to follow.

We thank the referee for the assessment that our work is good, that the new crystals achieved here are very high quality, and that the paper is well written.

I found however, many statements which are oversold and many other which I find myself in disagreement. The authors immediately state that the origin of the high conductivity is poorly understood. I disagree, there are several works, both theoretical and experimental, i.e., DFT, DMFT, Transport, quantum oscillations, ARPES, transport under irradiation etc., which have found quite solidly that the high conductivity of these systems is due to the s-dz orbitals hybridisation. This was, in agreement with the author conclusions attributed to Pd mobilities.

We have to respectfully disagree with this statement. There is no doubt that there are many prior works, both experimental and theoretical, that have contributed to the understanding that metallic delafossites are exceptional conductors. Transport measurements, including temperature-dependent studies, defect-density-dependent studies, quantum oscillation measurements, *etc.*, have clearly shown that the conductivity and mean-free-paths are staggeringly high. This establishes only that the scattering rates are very low, however, not *why* the scattering rates are so low, or *how* that can be, given the relatively dirty growth methods employed. This is exactly what our work is the *first* to fully achieve, showing that the overall purity is in fact not special, while the purity *in the Pd planes alone* is, enabling the very low point defect densities needed to attain this special transport. Similarly, theoretical works *via* DFT and DMFT have emphasized the electronic structure of these conductors, but cannot then address the key issue of *scattering rates*. Understanding of the latter is what our manuscript contributes for the first time, through the sublattice purification mechanism that we elaborate. This is the core message of our paper, which the other reviewers support.

Finally, with regard to the specific issue of *s-d* hybridization, please see the response to reviewer 1, point 3, including Figure R1.

On the one hand, the growth is genuinely impressive and the crystals obtained are, as the authors fairly report, much larger in size than the current ones. This is an excellent achievement. However, I do feel that the advance, compared to current existing and published literature is not enough to guarantee the publication in Nature Communications. I think the paper, which is a fine work, can find a more suitable fit in a more material oriented journal. At the moment, I do not see the advance to justify the fit with the current journal.

We thank the reviewer for the assessment that the crystal growth in our work is impressive, representing an excellent achievement. We think that the rest of this comment, however, does not fully acknowledge the main finding of our work, specifically the first elaboration of the sublattice purification mechanism that underpins the extraordinary conductivity of these metals, through fundamental principles of crystal chemistry. Prior work has established ultrahigh conductivity in these materials, and confirmed that this is rooted in very low defect density on the Pd planes, but our work is the *first* to understand *how* this occurs, establishing that the purity on the Pd planes greatly exceeds the average on the other atomic sites, and explaining this through crystal-chemical principles. As the other reviewers note, this is a definitive “answer” to this prominent problem.

Editorial requests

1. We have completed and uploaded the requested editorial policy checklist.
2. A “Data Availability” section is present in our manuscript. We have deposited our raw data in DRUM, the Data Repository for the University of Minnesota (<https://conservancy.umn.edu/drum>), with the following DOI: <https://doi.org/10.13020/0k3v-jj67>. This link is not yet live but will become so shortly.
3. The corresponding author has linked this work to their ORCID, and the other authors have also been asked to do so.

REVIEWERS' COMMENTS

Reviewer #1 (Remarks to the Author):

I found that the manuscript is satisfactorily revised. I recommend it for publication in Nature Communications.

Reviewer #2 (Remarks to the Author):

The authors have fully addressed my few comments from round 1, and I agree with their responses to the other referees. I am therefore happy to recommend publication.

Reviewer #3 (Remarks to the Author):

I am satisfied that the authors have dealt with my most substantive comment, in a way that improves the statistical basis of one of their key claims, that the samples are indeed different and of higher quality than those from other growth methods.

I recommend publication.

Reviewer #4 (Remarks to the Author):

I have carefully read the manuscript and I think the authors have convinced me about their findings. I think they have done an excellent job and provided the reviewers with very exhaustive answers.